



# Tracking vegetation phenology of pristine northern boreal peatlands by combining digital photography with CO₂ flux and remote sensing data

Maiju Linkosalmi[1], Juha-Pekka Tuovinen[1], Olli Nevalainen[1], Mikko Peltoniemi[3], Cemal M. Taniş[2], Ali N. Arslan[2], Juuso Rainne[1], Annalea Lohila[1], Tuomas Laurila[1], Mika Aurela[1]

[1]Finnish Meteorological Institute, Helsinki, Climate System Research, 00560 Helsinki, Finland
[2]Finnish Meteorological Institute, Helsinki, Arctic Space Centre, 00560 Helsinki, Finland
[3]Natural Resources Institute Finland (LUKE), 00790 Helsinki, Finland

*Correspondence to*: Maiju Linkosalmi (maiju.linkosalm@fmi.fi)

**Abstract.** Vegetation phenology, which refers to the seasonal changes in plant physiology, biomass and leaf area, is affected by many abiotic factors, such as precipitation, temperature and water availability. Phenology is also associated with the carbon dioxide (CO₂) exchange between ecosystems and the atmosphere. We employed digital cameras to monitor the vegetation phenology of three northern boreal peatlands during five growing seasons. We derived a greenness index (Green Chromatic Coordinate, GCC) from the images and combined the results with measurements of CO₂ flux, temperature and water table level, and with high-resolution satellite data (Sentinel-2). From the digital camera images it was possible to extract greenness dynamics on the vegetation community and even species level. The highest GCC and daily maximum gross photosynthetic production (GPP$_{max}$) were observed at the site with the highest nutrient availability and richest vegetation. The short-term temperature response of GCC depended on temperature and varied among the sites and months. Although the seasonal development and year-to-year variation of GCC and GPP$_{max}$ showed consistent patterns, the short-term variation in GPP$_{max}$ was explained by GCC only during limited periods. GCC clearly indicated the main phases of the growing season and peatland vegetation showed capability to fully compensate for the impaired growth resulting from a late growing season start. The GCC data derived from Sentinel-2 and digital cameras showed similar seasonal courses, but a reliable timing of different phenological phases depended upon the temporal coverage of satellite data.

## 1 Introduction

Boreal peatlands constitute a major terrestrial carbon (C) storage and continuously accumulate more C as a result of restricted decomposition of organic matter in anaerobic conditions. Boreal peatlands cover about 3% of the total land area, but they account for as much as a third of the global C pool (Gorham, 1991; Turunen et al., 2002). Climate and land use changes may disturb the functioning of these ecosystems and affect their exchange of carbon dioxide (CO₂) and other greenhouse gases (GHGs) with the atmosphere. Vegetation phenology, i.e. the seasonal changes in plant physiology, biomass and leaf area (Migliavacca et al., 2011; Sonnentag et al., 2011; Sonnentag et al., 2012; Bauerle et al., 2012), is one of the drivers of the C cycle of terrestrial ecosystems and is strongly linked to plant productivity and CO₂ exchange (Ahrends et al., 2009; Peichl et al., 2015; Toomey et al., 2015; Linkosalmi et al., 2016; Koebsch et al., 2019). Abiotic factors, such as precipitation, temperature, radiation and water availability, act as main drivers of ecosystem functioning and vegetation phenology (Bryant and Baird, 2003; Körner and Basler, 2010). Earlier onset of vegetation growth during the springtime, and thus a longer growing season, has been observed in recent decades in the boreal zone (Linkosalo, et al. 2009; Delbart et al., 2008; Nordli et al., 2008; Pudas et al., 2008). This strongly affects the annual C balance of ecosystems, because C accumulation starts as soon as environmental conditions become favourable for photosynthesis



and growth. In contrast, the corresponding lengthening in autumn does not have a similar effect (Goulden et al., 1996; Berninger, 1997; Black et al., 2000; Barr et al., 2007; Richardson et al., 2009), as also ecosystem respiration increases in late summer and autumn (White and Nemani, 2003; Dunn et al., 2007). In peatlands, the relationship between vegetation phenology and $CO_2$

exchange has been verified in several studies (Järveoja et al., 2018; Koebsch et al., 2019; Peichl et al., 2015; Peichl et al., 2018; Linkosalmi et al., 2016; Knox et al., 2017).

Remote sensing, both ground and satellite-based, is an effective tool for continuous monitoring of vegetation phenology and, thus, indirectly C fluxes. Time-lapse imaging with ground-based digital cameras provides small-scale information on the changes in the

vegetation observed, even on the species and vegetation community level. Several studies have shown that such repeat photography is capable of detecting the key patterns and events of vegetation phenology and it is possible to relate these observations to variations in $CO_2$ exchange (e.g. Wingate et al., 2015; Richardson et al., 2007; Richardson et al., 2009; Linkosalmi et al., 2016; Peichl et al., 2015; Koebsch et al., 2019). Especially, the Green Chromatic Coordinate (GCC) extracted from the red-green-blue (RGB) colour channel information of digital images has been used as an index of canopy greenness (e.g. Richardson et al., 2007;

Richardson et al., 2009; Ahrends et al., 2009; Ide et al., 2010; Sonnentag et al., 2012; Peichl et al., 2015; Peltoniemi et al., 2018).

Satellite data offer many benefits for land cover mapping: the data are cost-effective, cover large areas, and even the most remote sites are accessible (e.g. Lees et al., 2018). However, the vegetation and microtopography at many peatland sites are heterogeneous, which complicates the interpretation of satellite data. Furthermore, the presence of both vascular plants and mosses can be

challenging for satellite-based monitoring, as the species have different heights and cover each other forming an understory and other vegetation layers. The microtopography depends on the peatland type, and the surface can be relatively flat or patterned with strings and flarks.

In this study, vegetation phenology was observed with digital cameras in three natural peatlands in northern Finland for five

growing seasons. Our specific aims were to examine (1) how the GCC describes the variation of vegetation phenology between the sites and among different plant communities within one site, including the relationship between vegetation phenology and $CO_2$ flux dynamics, (2) how the abiotic factors (temperature, water table depth) modulate the development of GCC and $CO_2$ flux and (3) the potential use of satellite-derived GCC data for depicting the phenology of northern peatlands.

## 2 Material and methods

### 2.1 Sites

The three study sites are natural open peatlands, all located in northern Finland. Halssiaapa in Sodankylä (N67°22.117', E26°39.244', 180 m a.s.l.) is the southernmost of the sites and, as a mesotrophic fen, represents a typical aapa mire. The vegetation mainly consists of sedges (*Carex* spp.*),* big-leafed bogbean *(Menyanthes trifoliata),* bog-rosemary *(Andromeda polifolia),* dwarf birch *(Betula nana),* cranberry *(Vaccinium oxycoccos)* and peat moss *(Sphagnum* spp.). Tall trees are not present, only some minor

downy birch (*B. pubescens*) and Scots pine (*Pinus sylvestris*). Different types of vegetation are located on drier strings (shrubs) and wetter flarks (sedges and herbs). The trophic status varies from oligotrophic to eutrophic.

Lompolojänkkä (N67°59.842', E24°12.569', 269 m a.s.l.) is a nutrient-rich sedge fen located in the Pallas area. Of our study sites, Lompolojänkkä is the richest in nutrients. This is reflected in the vegetation, which is dominated by *B. nana, Menyanthes trifoliata,*





downy willow *(Salix lapponum)*, *Carex* spp. and *Sphagnum* and brown mosses (Aurela et al., 2009). Lompolojänkkä has the highest
leaf area index (one-sided LAI of 1.3 m$^2$ m$^{-2}$) (Aurela et al., 2009).

Kaamanen (N69°08.435', E27°16.189', 155 m a.s.l.) is the northernmost of the sites and also represents an aapa mire. The site is
located within the northern boreal vegetation zone, but the climate is already subarctic (Aurela et al., 1998). Vegetation is
distributed to wet flarks and strings of 0.3–0.6 m in height. On the strings, the vegetation mainly consists of ombrotrophic species,
such as forest mosses and Ericales (Maanavilja et al., 2011). *Sphagnum* mosses, sedges, *B. nana* and *Andromeda polifolia* dominate
the margins of the strings. The flarks are dominated by meso-eutrophic vegetation, such as brown mosses and sedges (Maanavilja
et al., 2011). Of these sites, the LAI is lowest (0.7 m$^2$ m$^{-2}$) at Kaamanen. At all sites, the snowmelt typically occurs in May.

### 2.2 Image analysis

The images were taken with StarDot Netcam SC 5 digital cameras and cover the years from 2015 to 2019. The cameras were
placed in a weatherproof housing and attached to line current and a remote web server. Images were taken automatically every 30
minutes with a 2592x1944 resolution in 8-bit JPEG format and transferred automatically to the server. The cameras were adjusted
in an angle of 45° on a pole facing the north. The cameras were mostly observing the peatland vegetation, but also the skyline was
visible in the images. The image quality settings (saturation, contrast and color balance) were the same in all cameras.


The data gained from the digital camera images consist of colour-based chromatic indices. The images were analyzed with the
FMIPROT (version 0.21.1) program that was designed as a toolbox for image processing for phenological and meteorological
purposes (Tanis et al., 2018). FMIPROT automatically derives the colour fraction indices from the images. We used the Green
Chromatic Coordinate (GCC):

$$GCC = \frac{\Sigma G}{\Sigma R + \Sigma G + \Sigma B}$$    (1)

where $\Sigma G$, $\Sigma R$, $\Sigma B$ are the sums of green, red and blue channel indices, respectively, of all pixels comprising an image. In FMIPROT
it is possible to choose different subareas, Regions of Interest (ROIs), within the image, for which GCC is calculated separately.
At the latitude of our study sites, solar radiation levels have been observed to be sufficient for image analysis from February to
October and the diurnal radiation levels acceptable from 11:00 to 15:00 (local winter time, +02:00 GMT) (Linkosalmi et al., 2016).
Here, we used images from the beginning of May to the end of September, which represent the growing season, and daily GCC
averages were calculated from the images taken during 11:00-15:00.

### 2.3 Regions of interest (ROIs)

The ROIs covering all different plant communities within the target area of the camera were defined for each site (Fig. 1 a-c). In
addition to these general ROIs, more specific ROIs (Fig. 1 d-f) were defined for clearly identifiable plant communities
characterized by specific dominant plant species (Table 1).





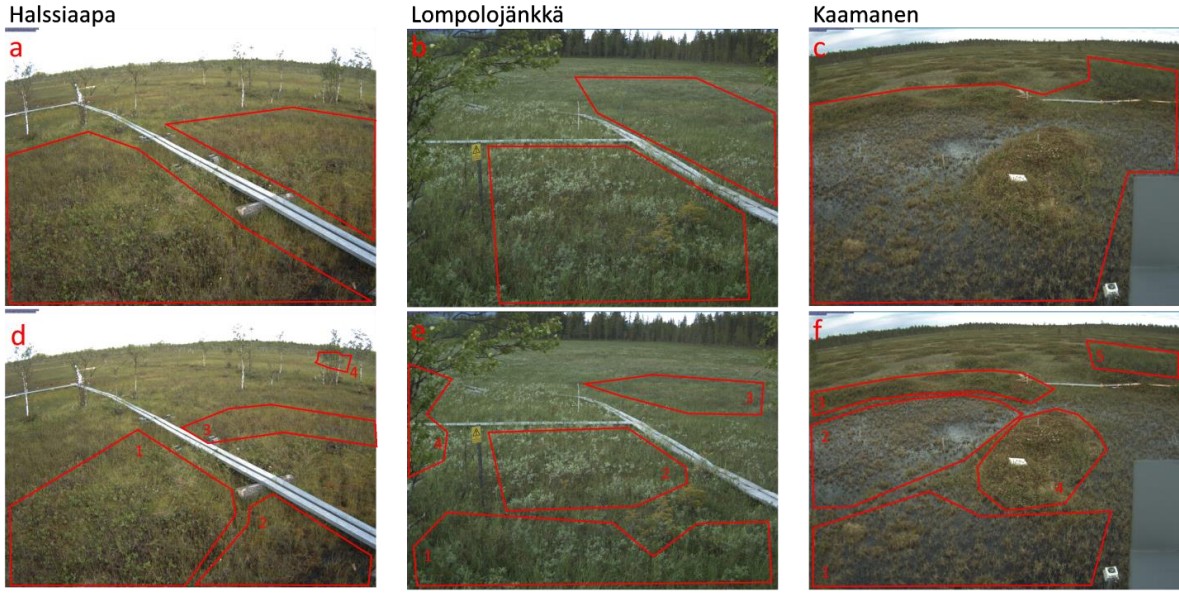

**Figure 1: The Region of Interests (ROIs) representing the overall vegetation at the site (a-c) and specific plant communities within the camera target areas (d-f). The numbers 1-5 indicate the plant communities detailed in Table 1.**

**Table 1: The dominant plant species characterizing the different plant communities within Region of Interests (ROIs) at**
**each site (Fig. 1 d-f).**

|      | Halssiaapa | Lompolojänkkä | Kaamanen |
|------|------------|---------------|----------|
| ROI1 | *Menyanthes trifoliata* | *Salix* sp., *Carex* spp. | *Carex* spp. and flark mosses |
| ROI2 | Sedges (mostly *Carex* spp.) | *Salix* sp., *Carex* spp. | *Carex* spp. and flark mosses |
| ROI3 | *Andromeda polifolia* and other shrubs | *Carex* spp. | *Empetrum nigrum, Rubus chamaemorus* |
| ROI4 | *Betula pubescens* | *Betula nana* | *Rhododrendron tomentusum, Rubus chamaemorus* |
| ROI5 |  |  | *Betula nana* |

**2.4 Ecosystem scale CO2 exchange and meteorological observations**

The ecosystem–atmosphere $CO_2$ exchange was measured by the micrometeorological eddy covariance (EC) method. The EC method provides continuous $CO_2$ flux data averaged on an ecosystem scale. The vertical $CO_2$ flux is defined as the covariance of
the high-frequency (10 Hz) fluctuations of vertical wind speed and $CO_2$ mixing ratio. At each site, the EC measurement system consisted of a USA-1 (METEK GmbH, Elmshorn, Germany) three-axis sonic anemometer and a closed-path LI-7000 (Li-Cor., Inc., Lincoln, NE, USA) $CO_2/H_2O$ gas analyzer. Air temperature, photosynthetic photon flux density (PPFD) and water table level were also measured at the sites. The measurement system and the data processing procedures have been presented in detail by Aurela et al. (2009).


The measured $CO_2$ flux represents the net ecosystem exchange (NEE), which is the sum of gross photosynthetic production (GPP) and ecosystem respiration. The daily maximum GPP, $GPP_{max}$, was calculated as the difference between the mean daytime (PPFD





$> 600\ \mu\text{mol m}^{-2}\ \text{s}^{-1}$) and nighttime (PPFD $< 20\ \mu\text{mol m}^{-2}\ \text{s}^{-1}$) NEE. The GPP$_{max}$ describes the seasonal GPP cycle and also reacts to short-term changes in air temperature and humidity (Aurela et al., 2001).

**2.5 Growing degree day sum, growing season start and temperature classes**

Growing Degree Day Sum (GDDS) was defined as the cumulative sum of the daily average temperatures exceeding 5 °C, each subtracted with the base value of 5 °C. The growing season was considered to start when the daily mean temperature has remained over 5 °C for ten days. The short-term change in GCC was expressed as a mean three-day difference, i.e. $\Delta$GCC = GCC(day$_t$)-GCC(day$_{t+3}$). It was calculated throughout the five growing seasons, and its monthly average was divided into three temperature

classes (<5 °C, 5-10 °C and >10 °C) calculated as a two-day average (day$_t$ and day$_{t+1}$). Also, cumulative GCC was calculated using the value just before the increase as the baseline. The cumulative sums were normalized by the maximum and minimum values of the year with the maximum cumulative GCC.

**2.6 Satellite data**

The GCC derived from the digital images of the ground-based cameras was compared with those derived from the Sentinel-2 data

acquired from 2016 to 2019. GCC was computed from the atmospherically corrected bottom-of-atmosphere products (Level-2A) using bands B2 (blue, 490 nm), B3 (green, 560 nm) and B4 (red, 665 nm) with a 10 m spatial resolution. Level-2A products were downloaded from the Sentinel Scientific Data Hub (https://scihub.copernicus.eu). If the Level-2A product was not available for a specific date, the Level-1C product was downloaded and processed to Level-2A product using the Sen2Cor software (version 2.8). Cloudy, cloud-shadowed and snowy satellite images were filtered using the scene classification data (SCL Band) available in the

Level-2A products.

GCC was calculated for multiple ROIs within each site (Fig. 2). These ROIs were different from those used with camera data because of the different spatial resolutions of the camera and satellite data. The selected ROIs represent different vegetation types with different microtopography within the study areas. The average of pixel-based GCCs within a ROI was used as the ROI-based

GCC. Site-based GCC was then calculated as the average of all ROI-based GCCs within the site. The Sentinel-2 images were available at the minimum for every two days, but the filtering reduced the number of valid images.

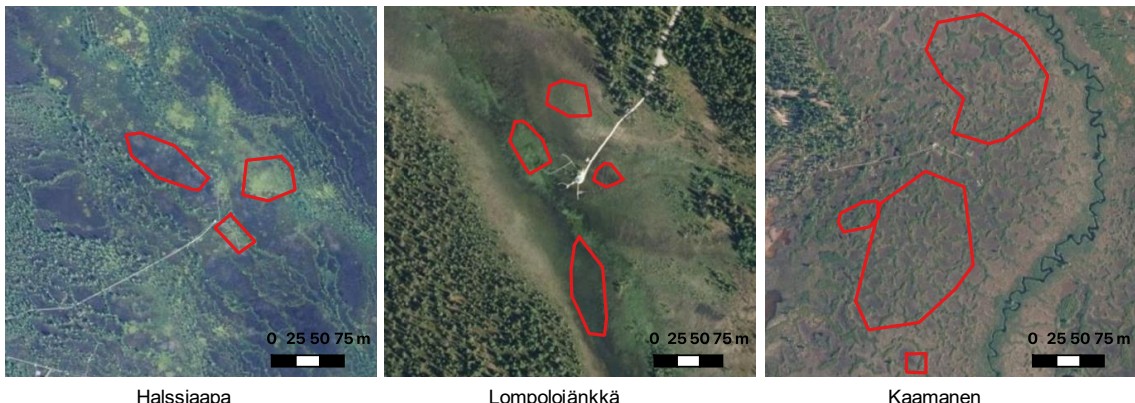

**Figure 2: The Region of Interests (ROIs) representing the overall vegetation at the sites for the Sentinel-2 satellite images.**
**The aerial photo contains data from the National Land Survey of Finland Topographic Database.**





### 2.7 Fitting of GCC and GPPmax cycles

To depict the phenology-driven seasonal cycle, we fitted a double hyperbolic tangent function to both camera- and satellite-derived GCC time series with the Levenberg-Marquardt least squares method (Meroni et al., 2014; Vrieling et al., 2018):

$$GCC(t) = a_0 + a_1 \frac{tanh\left((t-a_2)a_3\right)+1}{2} + a_4 \frac{tanh\left((t-a_5)a_6\right)+1}{2} - a_4, \tag{2}$$

where $t$ is time, $a_0$ is the minimum GCC value at the start of the growing season, $a_1$ ($a_4$) is the difference between the maximum GCC and minimum GCC, $a_2$ ($a_5$) is the inflection point in GCC development, and $a_3$ ($a_6$) controls the slope at the inflection point in GCC development during the first (second) half of the growing season. A similar function was fitted to GPP$_{max}$.

Visible snow included in the images affects the GCC data by overexposure. Thus, only the data collected after the snowmelt, which
usually occurs in May at all sites, was used. Also, the starting point of the fits was fixed to 1 May (Day of Year (DOY) 121), and the GCC value for this day was calculated as the average of the yearly minima after the snowmelt, whose timing was specified for each year and site. Likewise, the growing season ends by the end of October, and thus the end point of the fit was fixed to 31 October (DOY 304), for which GCC was determined by averaging annual minima in the end of the growing season.

From the fitted function, we calculated parameters that describe phenological phases and vegetation development. These parameters were chosen as the start of season (SOS25), which stands for 25% of the GCC difference between the maximum and 1 May, maximum GCC (MAX) and the end of season (EOS25), defined as 25% of the GCC difference between 30 October and the maximum.

### 2.8 Statistical analysis

The differences in GCC between the sites, different plant communities and measurement years were tested with the Kruskal-Wallis one-way analysis of variance on ranks for the group-wise comparison. Dunn's test was used for post-hoc testing. The Kruskal-Wallis method was used due to the non-normal distribution of the seasonal GCC data. The presence of autocorrelation in the residuals of the regression between GCC and GPP$_{max}$ was verified with the Durbin-Watson test. Autocorrelation was eliminated by regressing the first differences of the data, i.e. by applying the transformation $x'_t = x_t - x_{t-1}$ where $x_t$ and $x_{t-1}$ are consecutive
observations. The statistical analyses were performed with the R software (version 4.0.5).

## 3 Results

### 3.1 Greenness variation

The seasonal development of vegetation during the growing season could be visually observed from the imagery collected at our study sites, as exemplified by Figure 3. The spring development, greening and senescence of vegetation during the growing season
were visible in the images, and so were the changes in the areas covered by surface water.



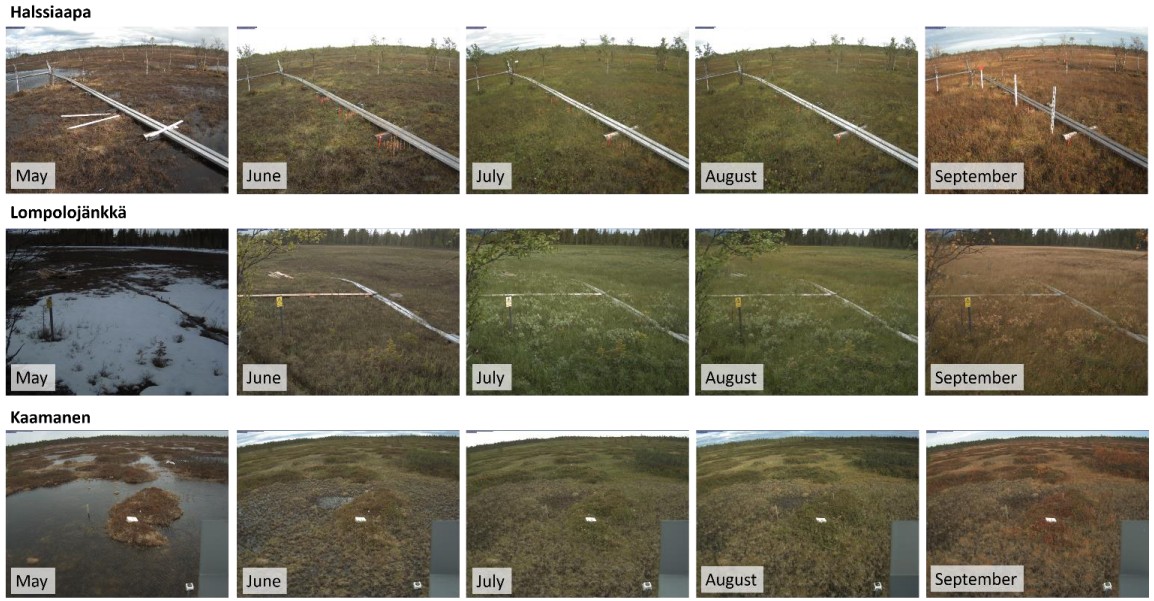

**Figure 3: The seasonal development of vegetation and surface water from May to September in 2015 at Halssiaapa, Lompolojänkkä and Kaamanen. The pictures were taken on the 15th of each month, with the exception of the 17th of June at Kaamanen.**

The mean growing season GCC values obtained from the phenological cameras showed that Lompolojänkkä systematically had the highest GCC values over the five growing seasons (Fig. 4 and Appendices Fig. A4). In 2015 and 2016, there was a significant difference ($p < 0.05$) in GCC between Lompolojänkkä and the other two sites. In 2017, 2018 and 2019, the pairwise comparison showed a significant ($p < 0.05$) difference in GCC between all the sites. The maximum GCC during the whole study period of 2015–2019 was observed in 2017 at Lompolojänkkä and Kaamanen and in 2016 at Halssiaapa (Fig. 4, Table 2).

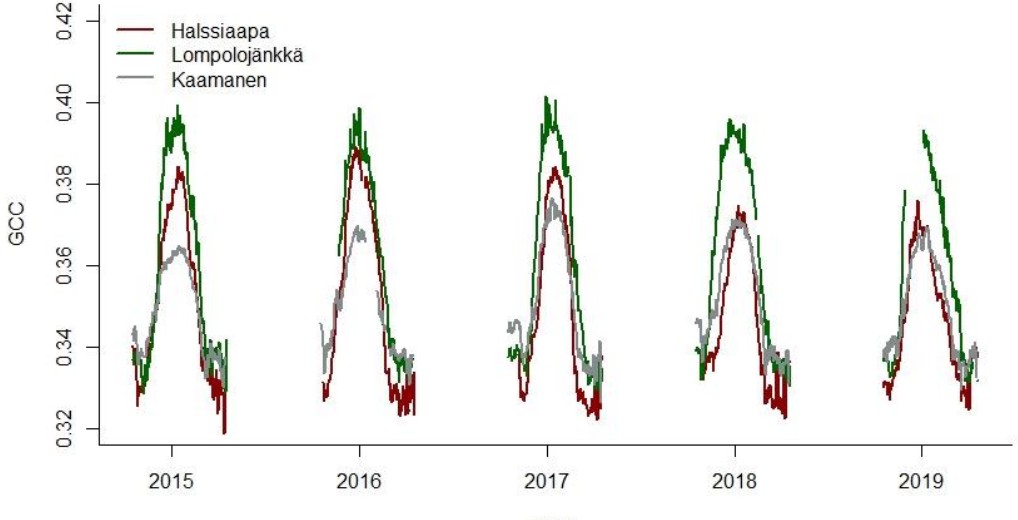

**Figure 4: Daily camera-based GCC values from 1 May to 30 September in 2015–2019.**





**Table 2: The mean GCC and GPPmax during the week they attained the maximum, the week numbers of these maxima and the week number of the growing season (GS) start. The GCC data from Lompolojänkkä and GPPmax data from Kaamanen from 2019 are missing.**

**Halssiaapa**

|  | Max GCC | Max GCC week | Max GP | Max GP week | GS start week |
|---|---|---|---|---|---|
| 2015 | 0.383 ± 0.018 | 31 | 0.318 ± 0.097 | 30 | 22 |
| 2016 | **0.388 ± 0.022** | **27** | **0.353 ± 0.108** | **27** | **18** |
| 2017 | 0.383 ± 0.020 | 31 | 0.338 ± 0.123 | 32 | 23 |
| 2018 | 0.371 ± 0.016 | 29 | 0.257 ± 0.090 | 32 | 19 |
| 2019 | 0.372 ± 0.015 | **27** | 0.302 ± 0.073 | 28 | 21 |

**Lompolojänkkä**

|  | Max GCC | Max GCC week | Max GP | Max GP week | GS start week |
|---|---|---|---|---|---|
| 2015 | 0.396 ± 0.024 | 31 | 0.520 ± 0.184 | 31 | 22 |
| 2016 | 0.394 ± 0.020 | **27** | 0.551 ± 0.185 | **27** | 20 |
| 2017 | **0.397 ± 0.024** | 29 | **0.563 ± 0.209** | 31 | 23 |
| 2018 | 0.394 ± 0.022 | **27** | 0.498 ± 0.181 | 30 | **19** |
| 2019 |  | - | 0.555 ± 0.197 | **29** | 22 |

**Kaamanen**

|  | Max GCC | Max GCC week | Max GP | Max GP week | GS start week |
|---|---|---|---|---|---|
| 2015 | 0.364 ± 0.010 | 31 | 0.299 ± 0.094 | 30 | 22 |
| 2016 | 0.368 ± 0.011 | **29** | 0.297 ± 0.089 | **28** | **19** |
| 2017 | **0.376 ± 0.014** | 30 | **0.335 ± 0.116** | 29 | 23 |
| 2018 | 0.371 ± 0.012 | **29** | 0.261 ± 0.090 | 32 | **19** |
| 2019 | 0.369 ± 0.012 | 30 | - | - | 22 |


There were significant GCC differences among different plant communities at all sites (p < 0.05), except in 2017 at Halssiaapa and Kaamanen (Appendices Table A1). In general, at all sites the GCC of birch species (*Betula pubescens* and *B. nana*) differed significantly from the other plant species. At Halssiaapa, the plant communities with sedges (*Carex spp.*) and shrubs (e.g. *Andromeda polifolia*) differed from annuals with bigger leaves, such as *Menyanthes trifoliata*. At Kaamanen, the shrubs and

annuals (e.g. *Empetrum nigrum*, *Rhododendron tomentusum*, *Rubus chamaemorus*) had a significantly higher GCC than the plant communities with sedges and flark mosses. The comparison of the maximum GCC values of different plant communities, calculated as weekly means, supported these results as the annuals and woody plants with relatively large leaves, such as *Menyanthes trifoliata*, *Rubus chamaemorus, Salix* spp. and *Betula spp.*, generated a higher GCC maximum than *Carex* spp. and shrubs (Table 3). At Halssiaapa, the highest maximum GCC during 2015–2017 and 2019 was observed in ROI1, which is

dominated by *Menyanthes trifoliata*, whereas in 2018 the highest GCC was found for ROI3, an area with *Andromeda polifolia* and other shrubs. At Lompolojänkkä, the highest annual GCC maximum was consistently observed in a plant community dominated by *Salix* sp. and *Carex* spp (ROI1). Most likely the ground layer with mosses and dead plant material reduced the GCC within those ROIs that had sparse vegetation. Among the measurement years, most plant communities at Halssiaapa showed the highest maximum GCC in 2016. At Lompolojänkkä, the maximum GCC of different ROIs varied between the years 2015, 2016 and 2017,

while at Kaamanen all plant communities attained their maxima in 2017.





**Table 3: The maximum GCC values of different plant communities characterized by the dominant species specified in Table 1 (defined as image ROIs) from 2015 to 2019. The maximum value among different ROIs is marked as bold, and the maximum among the years is underlined. The ROIs are described in Section 2.4. The data from Lompolojänkkä are missing in 2019.**

**Halssiaapa**

|      | ROI1 | ROI2 | ROI3 | ROI4 |
|------|------|------|------|------|
| 2015 | **0.387 ± 0.019** | 0.373 ± 0.016 | 0.376 ± 0.018 | 0.372 ± 0.016 |
| 2016 | **_0.391 ± 0.023_** | _0.378 ± 0.019_ | _0.382 ± 0.021_ | 0.371 ± 0.015 |
| 2017 | **0.384 ± 0.020** | 0.358 ± 0.018 | 0.381 ± 0.021 | _0.372 ± 0.016_ |
| 2018 | 0.370 ± 0.015 | 0.365 ± 0.014 | **0.374 ± 0.018** | 0.370 ± 0.013 |
| 2019 | **0.374 ± 0.015** | 0.366 ± 0.013 | 0.368 ± 0.014 | 0.371 ± 0.013 |

**Lompolojänkkä**

|      | ROI1 | ROI2 | ROI3 | ROI4 |
|------|------|------|------|------|
| 2015 | **_0.406 ± 0.027_** | _0.395 ± 0.024_ | 0.388 ± 0.022 | _0.403 ± 0.026_ |
| 2016 | **0.400 ± 0.019** | 0.393 ± 0.022 | _0.396 ± 0.021_ | 0.397 ± 0.024 |
| 2017 | **_0.406 ± 0.027_** | 0.393 ± 0.023 | _0.396 ± 0.023_ | 0.402 ± 0.027 |
| 2018 | **0.401 ± 0.023** | 0.392 ± 0.022 | 0.393 ± 0.022 | 0.400 ± 0.024 |
| 2019 | - | - | - | - |

**Kaamanen**

|      | ROI1 | ROI2 | ROI3 | ROI4 | ROI5 |
|------|------|------|------|------|------|
| 2015 | 0.358 ± 0.008 | 0.357 ± 0.007 | **0.376 ± 0.016** | 0.375 ± 0.016 | 0.374 ± 0.019 |
| 2016 | 0.363 ± 0.009 | 0.362 ± 0.008 | **0.380 ± 0.016** | 0.379 ± 0.016 | 0.377 ± 0.017 |
| 2017 | _0.371 ± 0.011_ | _0.368 ± 0.010_ | _0.384 ± 0.019_ | **_0.388 ± 0.021_** | _0.385 ± 0.019_ |
| 2018 | 0.365 ± 0.009 | 0.365 ± 0.009 | 0.379 ± 0.015 | **0.381 ± 0.017** | 0.378 ± 0.016 |
| 2019 | 0.363 ± 0.008 | 0.362 ± 0.007 | **0.379 ± 0.015** | 0.377 ± 0.015 | 0.378 ± 0.016 |

### 3.2 Temperature and GCC development

The relationship between temperature and GCC was examined by creating normalized cumulative GCC and GDDS curves for all the growing seasons (Appendices Fig. A4). The cumulative sums show that the GCC started to accumulate later than GDDS. In 2017, the snow cover lasted at all sites until the beginning of June, which delayed the GDDS development and the start of the growing season compared to the other study years (Figs. 5 and 7 and Appendices Figs. A3 and A4). Consequently, GCC did not increase until the beginning of June (4 June in Halssiaapa and Lompolojänkkä and 7 June in Kaamanen), when the snow was melted. Despite the slow start of growth in 2017, the peatland vegetation was capable of catching up its typical development, and at Lompolojänkkä and Kaamanen GCC even reached the highest summer maximum during the study years (Table 2).

During the measurement years, warmer springs and thus earlier snowmelts resulted in an earlier green-up of vegetation. No clear connection between the growing season start and the timing of the maximum of either GCC or $GPP_{max}$ was found at Lompolojänkkä or Kaamanen (Table 2). At Halssiaapa, however, the earliest growing season start, the maximum value and the earliest timing of both GCC and $GPP_{max}$ occurred in the same year, 2016. During our study period, the year 2018 had the warmest summer at all sites (Appendices Figs. A2 & A3). At Halssiaapa, the water table depth increased substantially during the growing seasons of 2018 and 2019 as a result of drought (Appendices Fig. A2). The same was true at Kaamanen (data missing in 2019), while at Lompolojänkkä WTD was not affected. The effect of drought is also visible in the daily and cumulative GCC values (Fig. 4 and Appendices Fig. A4).



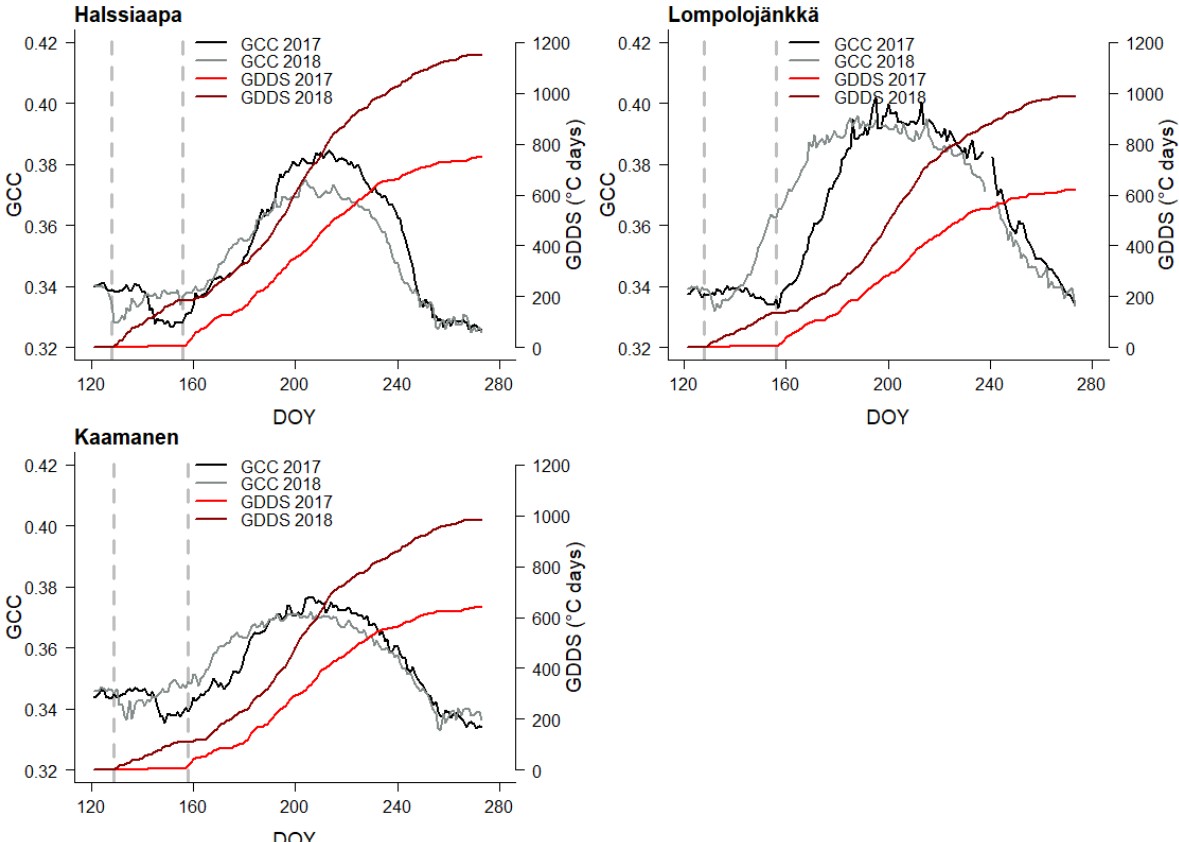


**Figure 5: Development of GCC and growing degree day sum (GDDS) from 1 May to 30 September at Halssiaapa, Lompolojänkkä and Kaamanen. The years 2017 and 2018 are shown here as an example of a cold and warm spring, respectively. The grey dashed lines indicate the start of the growing season.**

The short-term GCC change ΔGCC, expressed here as the monthly mean 3-day difference, depended on both the month and temperature range (Fig. 6). In May, this change, which is indicative of vegetation development, was substantially smaller for temperatures below than above 5 °C. No significant ($p < 0.05$) differences were found between the sites, nor during any month in the lowest temperature class (Fig. 6). At Lompolojänkkä, ΔGCC started at a lower temperature and was generally larger than at the other sites. The vegetation growth in June at Halssiaapa seemed to benefit from temperatures over 10 °C, while at

Lompolojänkkä this limit was lower. At Kaamanen, however, the ΔGCC in June was similar in all temperature classes. In July, GCC started to stabilize and a significant positive change was only observed at Kaamanen for temperatures between 5 and 10 °C and at Halssiaapa with temperatures over 10 °C. In August and September, ΔGCC was negative due to senescence (Fig. 6). The ΔGCC of different plant communities (Appendices Fig. A5) showed that the growth of *Betula spp.* started strong in May at all sites, but in June the birches already had a lower ΔGCC than other plant communities. The highest plant community-specific

ΔGCC values were found at Lompolojänkkä, which was consistent with the spatially averaged ΔGCC data (Figs. 4 & 6).







**Figure 6: Mean three-day difference in GCC divided to temperature classes (<5 °C, 5-10 °C, >10 °C) at Halssiaapa (Hal), Lompolojänkkä (Lom) and Kaamanen (Kaa) from May to September. There are no temperature data in the <5 °C class in July and August. The error bars denote the standard error, and the asterisks denote the statistically significant (p<0.05) difference between the sites.**






### 3.3 Seasonal GCC and GPPmax development

The start and end of the growing season were clearly visible in both the GCC and GPP$_{max}$ data, which showed the same seasonal pattern (Figs. 7 & 8). As mentioned above, in 2017 the snow cover lasted until the beginning of June, which delayed the start of photosynthesis and thus vegetation development. At Halssiaapa, the year 2018 was hot and dry (Appendices Fig. A2), and this was

reflected in the GCC and GPP$_{max}$ data that were lower than in other years (Fig. 7 & 8). The GPP$_{max}$ data were sparse during the growing season of 2018, which impaired the fit (Fig. 8). During the study period, the highest values of both GCC and GPP$_{max}$ were observed in 2016 at Halssiaapa and in 2017 at Kaamanen and Lompolojänkkä (Figs. 7 & 8). In general, Lompolojänkkä had the highest GPP$_{max}$, GCC and LAI (Figs. 4, 7 and 8).

The difference in GCC was significant ($p < 0.05$) between Lompolojänkkä and the other two sites in all years, and in 2017, 2018 and 2019 also between Halssiaapa and Kaamanen. In 2015, GPP$_{max}$ showed no difference among the sites, but in 2016 and 2019 there was a significant difference between Kaamanen and the other two sites, and in 2017 and 2018 between Lompolojänkkä and Kaamanen. The GPP$_{max}$ and GCC showed a similar course throughout the growing season (Appendices Fig. A9).

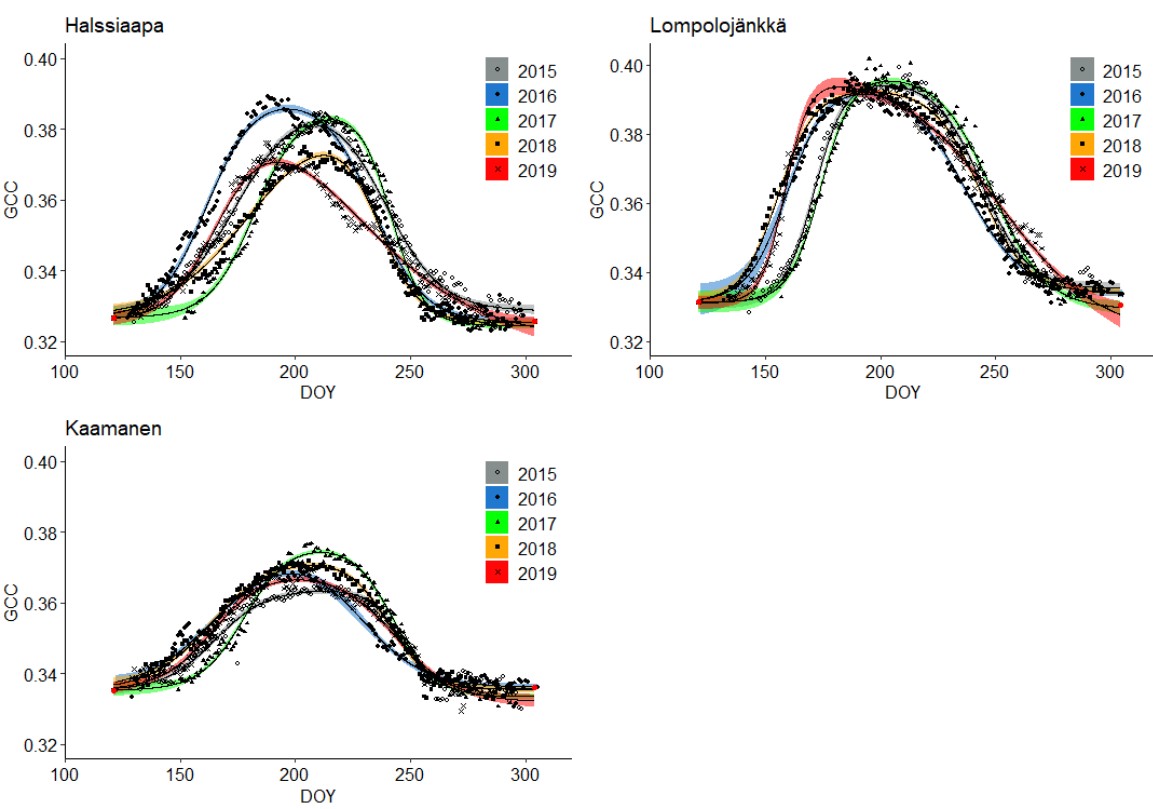


**Figure 7: Development of GCC from 1 May to 30 September in 2015–2019 at Halssiaapa, Lompolojänkkä and Kaamanen. Different symbols and colours denote different years, and the bands show the 95% confidence intervals of the fitted double hyperbolic tangent function. The red dots indicate the fixed start and end days defined for the fitting.**





**Figure 8: Development of GPPmax from 1 May to 30 September in 2015–2019 at Halssiaapa, Lompolojänkkä and Kaamanen. Different symbols and colours denote different years, and the bands show the 95% confidence intervals of the fitted double hyperbolic tangent function. The red dots indicate the fixed start and end days defined for the fitting. Note the differences in y-axis scaling.**

A linear relationship between GCC and $GPP_{max}$ was observed during both the increasing and decreasing phases of GCC (Table 4, Appendices Fig. A8). However, the Durbin-Watson test indicated that there was significant and strong autocorrelation in the model residuals. After differencing the data, the coefficient of determination was generally close to zero. There were periods showing correlated short-term variation in GCC and $GPP_{max}$, for example, at Kaamanen from late May to mid-June in 2016 and in June 2017 (Appendices Fig. A9 c), but most of the common variation in $GPP_{max}$ and GCC was associated with the common seasonal cycle (Table 4).





**Table 4: The coefficient of determination ($R^2$) of the linear regression between GCC and GPPmax and of the regression after differencing the data for autocorrelation. Letters a and b refer to the period before and after the annual GCC maximum, respectively.**

| | Halssiaapa | | Lompolojänkkä | | Kaamanen | |
|---|---|---|---|---|---|---|
| | | Autocorr. | | Autocorr. | | Autocorr. |
| | Original $R^2$ | corr. $R^2$ | Original $R^2$ | corr. $R^2$ | Original $R^2$ | corr. $R^2$ |
| 2015 a | 0.9749 | 0.0291 | 0.9549 | 0.0055 | 0.9372 | 0.0002 |
| 2015 b | 0.9305 | 0.0015 | 0.9697 | 0.0082 | 0.9134 | 0.0023 |
| 2016 a | 0.9848 | 0.0020 | 0.8668 | 0.1096 | 0.9304 | 0.0002 |
| 2016 b | 0.9660 | 0.0112 | 0.9686 | 0.0001 | 0.8749 | 0.0001 |
| 2017 a | 0.9448 | 0.0133 | 0.9599 | 0.3100 | 0.9171 | 0.0195 |
| 2017 b | 0.9634 | 0.0550 | 0.9697 | 0.0138 | 0.9724 | 0.0065 |
| 2018 a | 0.4468 | 0.0860 | 0.8890 | 0.1413 | 0.8944 | 0.0101 |
| 2018 b | 0.9388 | 0.0088 | 0.9840 | 0.0119 | 0.9029 | 0.0045 |
| 2019 a | 0.8513 | 0.0473 | 0.9162 | 0.0124 | 0.9854 | 0.1621 |
| 2019 b | 0.8198 | 0.0024 | 0.9621 | 0.0011 | 0.9394 | 0.0570 |

### 3.4 Comparison between digital camera- and satellite-derived GCC

The retrieved GCC from the Sentinel-2 images had the same seasonal pattern as the camera-derived GCC (Fig. 9). Due to the sparseness of satellite data, however, the uncertainties were greater, as shown by the wider confidence intervals. The later season start and GCC maximum in 2017 at all sites and the lower GCC at Halssiaapa in 2018, compared to the other measurement years that were observed with cameras, were visible in the satellite-derived GCC. The GCC values from Sentinel-2 were in general higher than the camera-based GCC, which is most probably due to the different viewing angles and atmospheric effect (the scattering and absorption of radiation due to atmospheric molecules and aerosols) and the consequent atmospheric correction of the satellite data.





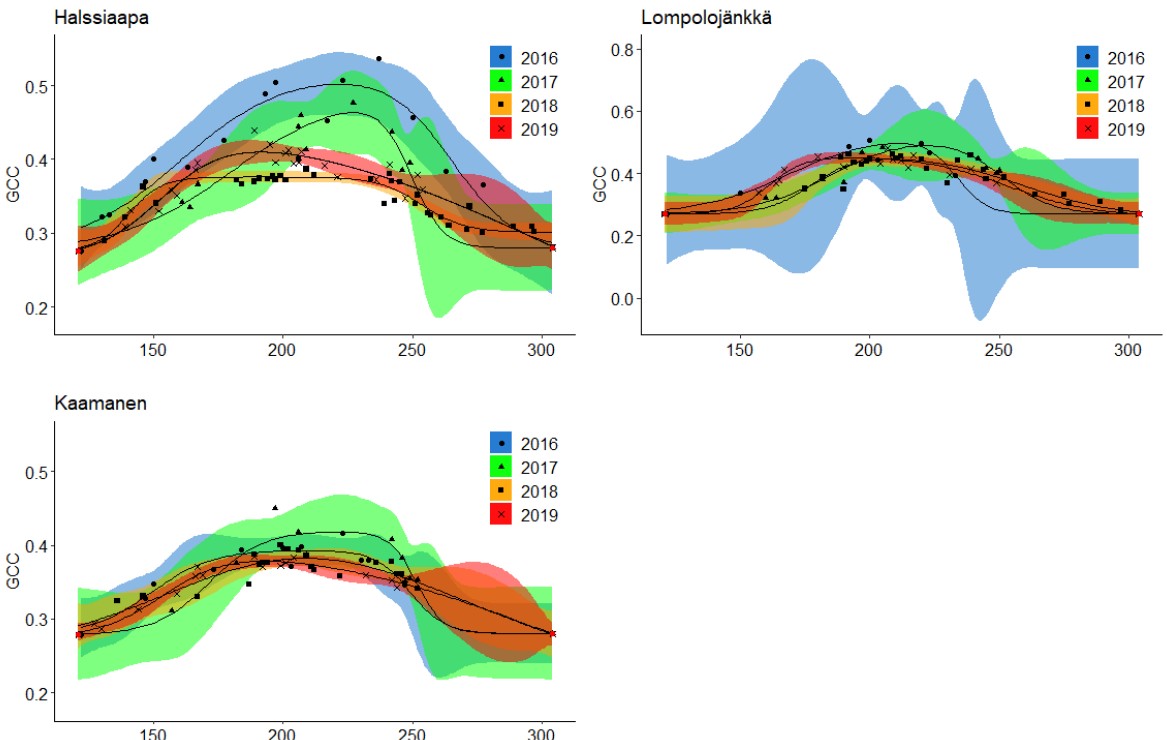

**Figure 9: Mean GCC derived from the Sentinel-2 data from 1 May to 30 September in 2016–2019 at Halssiaapa,**
305 **Lompolojänkkä and Kaamanen. Different symbols and colours denote different years, and the bands show the 95%**
**confidence intervals of the fitted double hyperbolic tangent function. The red dots indicate the fixed start and end days**
**defined for the fitting.**

The estimated parameters describing different growing season phases, which were calculated from the fitted models, Eq. (2), show

substantial differences between satellite and camera data (Table 4). At Halssiaapa and Kaamanen, the season start (SOS25) was

310 estimated to start earlier based on the Sentinel-2 data, whereas at Lompolojänkkä the timing of SOS25 and MAX occurred later.

The timing of the end of season (EOS25) was estimated later with the Sentinel-2 data at Halssiaapa and Kaamanen, while at

Lompolojänkkä there was no systematic difference.





**Table 5: Growing season phases estimated from camera and satellite images (DOY): SOS25 (Start of season, 25% difference between start and maximum), MAX (maximum GCC) and EOS (End of season, 25% difference between end and maximum).**

**Halssiaapa**

| | SOS25 | | | Max | | | EOS25 | |
|---|---|---|---|---|---|---|---|---|
| | Camera | Satellite | | Camera | Satellite | | Camera | Satellite |
| 2015 | 157 | | 2015 | 208 | | 2015 | 257 | |
| 2016 | 150 | 145 | 2016 | 197 | 222 | 2016 | 249 | 280 |
| 2017 | 170 | 157 | 2017 | 215 | 227 | 2017 | 251 | 258 |
| 2018 | 157 | 133 | 2018 | 212 | 191 | 2018 | 250 | 265 |
| 2019 | 152 | 139 | 2019 | 192 | 191 | 2019 | 265 | 288 |

**Lompolojänkkä**

| | SOS25 | | | Max | | | EOS25 | |
|---|---|---|---|---|---|---|---|---|
| | Camera | Satellite | | Camera | Satellite | | Camera | Satellite |
| 2015 | 157 | | 2015 | 202 | | 2015 | 258 | |
| 2016 | 147 | 151 | 2016 | 193 | 211 | 2016 | 255 | 241 |
| 2017 | 168 | 167 | 2017 | 204 | 222 | 2017 | 264 | 265 |
| 2018 | 153 | 166 | 2018 | 196 | 208 | 2018 | 257 | 279 |
| 2019 | 149 | 152 | 2019 | 182 | 193 | 2019 | 274 | 273 |

**Kaamanen**

| | SOS25 | | | Max | | | EOS25 | |
|---|---|---|---|---|---|---|---|---|
| | Camera | Satellite | | Camera | Satellite | | Camera | Satellite |
| 2015 | 153 | | 2015 | 213 | | 2015 | 255 | |
| 2016 | 147 | 139 | 2016 | 198 | 210 | 2016 | 250 | 258 |
| 2017 | 166 | 160 | 2017 | 211 | 222 | 2017 | 255 | 258 |
| 2018 | 149 | 139 | 2018 | 207 | 209 | 2018 | 253 | 288 |
| 2019 | 151 | 140 | 2019 | 201 | 195 | 2019 | 261 | 288 |

## 4 Discussion

In this study, we examined how phenology, described with GCC, varied in three natural peatlands in northern Finland during five growing seasons and how it depends on the site-specific characteristics and the composition of vegetation. The collected data allowed us to create a continuous representation of the development of greenness, which could be related to observed changes in the ecosystem-atmosphere flux of $CO_2$ and compared with the corresponding satellite-derived data.

We found the highest GCC values at Lompolojänkkä, a fen with high nutrient availability, rich vegetation and the highest LAI of the study sites (Fig. 4). The highest GCC also coincided with the highest photosynthetic productivity at peak summer. At Lompolojänkkä, the surface is flatter than at the other sites, where pronounced microtopography results in a higher variability in the hydrological features and consequently in the trophic status and vegetation. Also, the stream running through the Lompolojänkkä fen feeds water and nutrients to its surroundings (Lohila et al. 2010, Aurela et al. 2009). This affects fen's nutrient status and is reflected in vegetation, which mainly consists of annuals such as *Carex* spp. and *Menyanthes trifoliata,* and thus in the GCC and $GPP_{max}$ reported here.



Related to the microtopographical differences between the study sites, we found a higher GCC in those plant community types that had annuals with bigger leaves and, correspondingly, a lower GCC in areas dominated by shrubs and mosses (Table 3, Appendices Fig. A1). The different plant communities also have different habitats. For example, shrubs thriving in drier locations such as
strings, while many annuals (sedges, big-leaved bogbean) favour wetter environment.

Compared to traditional and more laborious measurements of plant growth, the digital camera-based measurements produce automatically high-frequency data in an effortless way. Even though the variation among the sites was greater than the variation between the ROIs within one site, our results indicate that vegetation monitoring is feasible even at the plant community level. To
our knowledge, different plant communities have not been defined and compared from digital camera images with this precision before, and we conclude that these GCC observations of the differentially developing vegetation types have a potential to help decomposing an integrated $CO_2$ flux observation into components allocated to these vegetation types.

A warm spring leads to an earlier growing season start. Nevertheless, even though the growing season start was late at the study
sites in 2017, due to a cold spring and late snow melt, vegetation was capable of reaching the same maximum GCC level as in other years, at Lompolojänkkä and Kaamanen even attaining the maximum GCC and $GPP_{max}$ observed during the whole study period (Fig. 5). A review of Wipf and Rixen (2010) on arctic and alpine ecosystems concluded that a delayed snowmelt and thus a shorter growing season decreases the overall plant productivity of an ecosystem, but it also noted that the effect of snowmelt timing depends on the plant functional type; for example, the growth of forbs increases while the growth of grasses decreases when
the snowmelt occurs later. The later phenological phases are most likely controlled by GDDS rather than the timing of the growing season start (Wipf, 2010). Furthermore, the phenology of those plant species which usually start developing earlier after the snowmelt, and the first phenological phases of all plant species, are more sensitive to changes in the snowmelt timing. Our results imply that the northern peatland vegetation is capable of starting the growth quickly after a cold spring and the vegetation can even increase gross primary production, if the conditions are favourable later during the growing season, as was the case in 2017. This
was clearly observed in the magnitude and timing of the maximum GCC. The faster GCC increase and lower temperature sensitivity at Lompolojänkkä than at the other sites are explained by the nutrient status of this fen.

As observed in several studies conducted in different ecosystems (forests, grasslands, crops, peatlands), GPP correlates strongly with the greenness index derived from digital camera images (Richardson et al., 2008; Migliavacca et al., 2011; Keenan et al.,
2014; Peichl et al., 2015; Toomey et al., 2015; Linkosalmi et al., 2016; Knox et al., 2017; Järveoja et al., 2018; Peichl et al., 2018; Koebsch et al., 2019). Our results agree with these studies, showing highly similar seasonal cycles of GCC and $GPP_{max}$ at open peatlands dominated by shrubs and deciduous plants. Essentially, both are controlled by the amount of green leaf area, which in turn is driven mainly by temperature and day length (Bauerle et al., 2012; Peichl et al., 2015; Koebsch et al., 2019). When temperature increases, the plant chemical reaction rates also increase, triggering photosynthesis (Bonan, 2015). According to our
results, air temperature, expressed here as a degree day sum (Fig. 5), explained well the annual differences in the early phase of the growing season, which has also been previously observed for a more southern boreal peatland (Peichl et al. 2015). In the latter part of the growing season, the decreasing day length and temperatures strongly drive the gradual degradation of chlorophyll content, which eventually leads to downregulation of photosynthesis and further to leaf fall and winter dormancy (Larcher, 2003; Öquist and Huner, 2003; Bonan, 2015).




While the differences in the phenological courses during different years are well depicted by GCC, also the variation in the maximum greenness level among the years and sites may be assessed by the phenocamera-derived GCC. Lompolojänkkä showed during all the years significantly higher maximum GCC and $GPP_{max}$ than the other sites (Figs. 7 & 8, Table 2). Also, the maximum GCC and $GPP_{max}$ were found in the same year at all sites. In addition to the seasonal cycles, both GCC and GPP show distinct

periods of correlated short-term variation, which is mainly controlled by abiotic factors, such as temperature and solar radiation (Peichl et al., 2015). For example, the variation in GCC and $GPP_{max}$ was highly similar during the first part of the growing seasons of 2016 and 2017 at Kaamanen (Appendices Fig. A9 c) and during the drought period in 2018 at Halssiaapa (Figs. 7 & 8, Appendices Fig. A6). At Lompolojänkkä, this widespread drought did not result in a decrease of the water table level, which was due to the local hydrological features, and the net $CO_2$ uptake even increased in contrast to other northern sites (Rinne et al., 2010).

At Kaamanen, however, the drought decreased the $CO_2$ sink, counterbalancing the gain due to the earlier growing season start in 2018 (Heiskanen et al., 2021). Our data show that the drought in 2018 and 2019 affected the $CO_2$ sink most at Halssiaapa (Fig. 8, Appendices Fig. A7). The greenness data, both from the Sentinel-2 satellite and repeat photography, are in good accord with these observations of $CO_2$ flux dynamics (Figs. 7 & 9). Despite the specific periods of correlation, our regression analysis indicates that most of the variation in $GPP_{max}$ can be explained by the common seasonal cycle, rather than the short-term variations of GCC

(Table 4).

The fitting of a hyperbolic tangent function to GCC data to characterize the basic phenological cycle worked well when data availability was sufficient (Figs. 7 & 9). The effect of poor data coverage is especially evident in the Sentinel-2 data in the beginning and end of the growing season, and also when the camera data were limited, e.g. in the early growing season at Halssiaapa in 2018

and at Kaamanen in 2019. Such data losses obviously compromise the accurate timing of the phenological phases. We also found large differences between the camera- and satellite-derived growing season phases (Table 4), the fits to the Sentinel-2 data suggesting a longer growing season at Halssiaapa and Kaamanen. Nevertheless, the main phenological changes during the growing season were visible also in the satellite data. Vrieling et al. (2018) found large differences between the phenological parameters derived from the satellite-based NDVI and camera-based GCC time series, especially in the end of the growing season. The

differences could be explained by the non-photosynthetic vegetation mass, such as dead plant matter, stems and flowers, which in an angled view affects the visibility of the green plant mass in the image. Thus, Vrieling et al. (2018) suggested that camera observations taken at nadir, rather than from an angled view, could produce better correlation between satellite and camera data. Also, it should be noted that the satellite indices are estimated from surface reflectance, while the camera image analysis applies raw digital numbers that scale with the reflected radiance (Vrieling et al., 2018). Thus, the resulting GCC estimates can be expected

to differ between the techniques, as also shown by the present study.

Obviously, the temporal coverage of non-cloudy satellite data, typically including an image every 5 to 10 days, is limited compared to the high-frequency camera-based measurements. The satellite data were limited especially during 2016 and 2017, because at that time Sentinel-2 constellation (Sentinel-2A and Sentinel-2B) consisted only of Sentinel-2A. Since 2018, however, there have

been data available from two satellites. Overall, mapping the vegetation on these heterogeneous peatlands with remote sensing methods is challenging and the suitability of the methods depends on the peatland structure (Räsänen et al., 2019). By providing local, continuous data even on the plant community level, digital photography could be used for verification of remote sensing products and as supporting information for their interpretation, as well as for filling the gaps in the landscape level data (Filippa et al., 2018; Richardson et al., 2007; Richardson et al., 2009; Sonnentag et al., 2012).





**5 Conclusions**

In this study, we showed that the digital photography derived greenness index (GCC) differed between three northern boreal peatland sites, being associated with nutrient availability and LAI. At all sites, the seasonal course of GCC was closely correlated with that of $CO_2$ uptake. The digital images also enabled determining the GCC of different plant communities, suggesting that these images can potentially be used for partitioning the ecosystem-scale $CO_2$ flux measurement. The spring temperatures and

consequent variation in growing season start affected GCC and $GPP_{max}$, but the peatland vegetation showed capability to compensate for a late start, and even to reach the maximum growth level observed during the five study years. The effect of drought on GCC and $GPP_{max}$ depends on local hydrological features and thus the drought resistance of the site. Despite the seasonal coherence between the GCC and $GPP_{max}$ data, the short-term variation of GCC did not in general explain the corresponding variation in $GPP_{max}$. The remote sensing (Sentinel-2) data were consistent with the camera-based results, but more satellite data

would be needed for a more reliable timing of different phenological phases. Finally, we conclude that the digital photography data could be used for verification, interpretation and gap filling of the remote sensing data.



**Appendices**

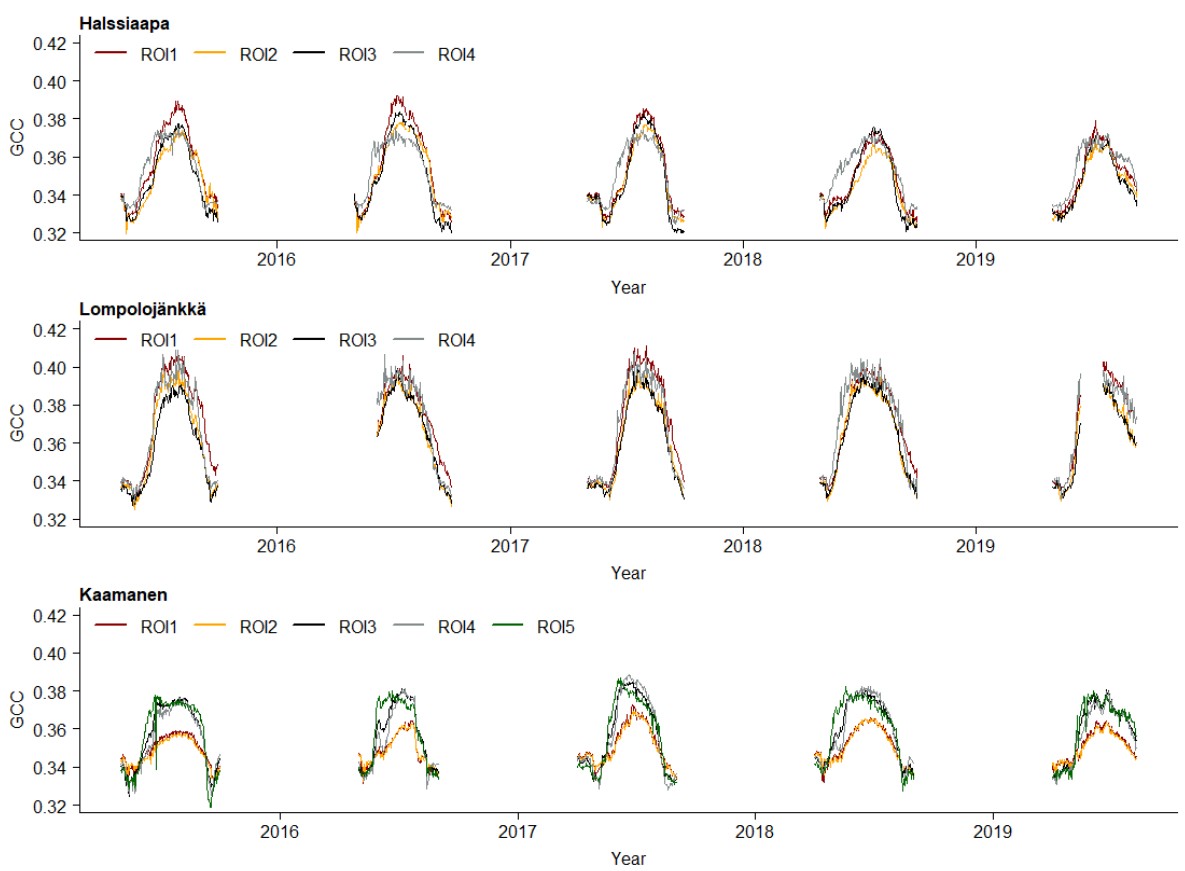

**Figure A1: The GCC of the different Region of Interests (ROIs) of specific plant communities. The numbers 1-5 indicate**
**the different plant communities detailed in Table 1.**

**Table A1: The significant differences between different plant communities. The numbers 1-5 indicate the different plant communities detailed in Table 1. For interpretation, a denotes significant (p< 0.05) difference from ROI1, b denotes significant difference from ROI2, c denotes significant difference from ROI3, d denotes significant difference from ROI4**
**and e denotes significant difference from ROI5.**

|  | 2015 | 2016 | 2017 | 2018 | 2019 |
|---|---|---|---|---|---|
| **Halssiaapa** | ROI2 a | ROI3 a |  | ROI2 a | ROI2 a, d |
|  | ROI3 a, d |  |  | ROI3 a, b, c | ROI3 a, d |
|  |  |  |  |  |  |
| **Lompolojänkkä** | ROI2 a, d | ROI2 a, d | ROI2 a, d | ROI2 a, d | ROI2 a, d |
|  | ROI3 a, d | ROI3 a, d | ROI3 a, d | ROI3 a, d | ROI3 a, d |
|  |  |  |  |  |  |
| **Kaamanen** | ROI1 c, e | ROI1 c, e |  |  | ROI1 c, d, e |
|  | ROI2 c, d, e | ROI2 c, e |  |  | ROI2 c, d, e |



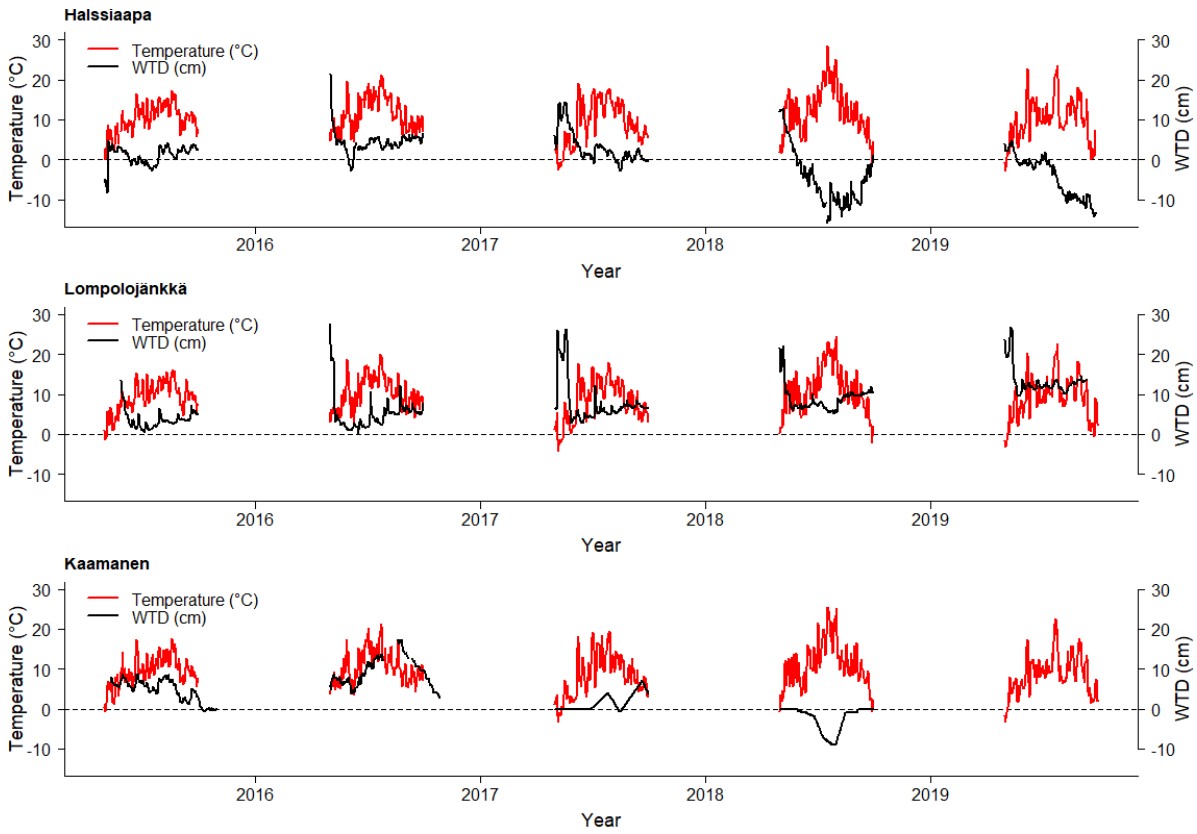

Figure A2: Daily mean temperatures (°C) and water table depths (cm) in 2015–2019 at the experimental sites.





**Figure A3: The growing degree days (GDDS) in 2015 – 2019 at the experimental sites.**



**Figure A4 a: The normalized cumulative GCC and growing degree days (GDDS) in 2015 – 2019 at the experimental sites.**



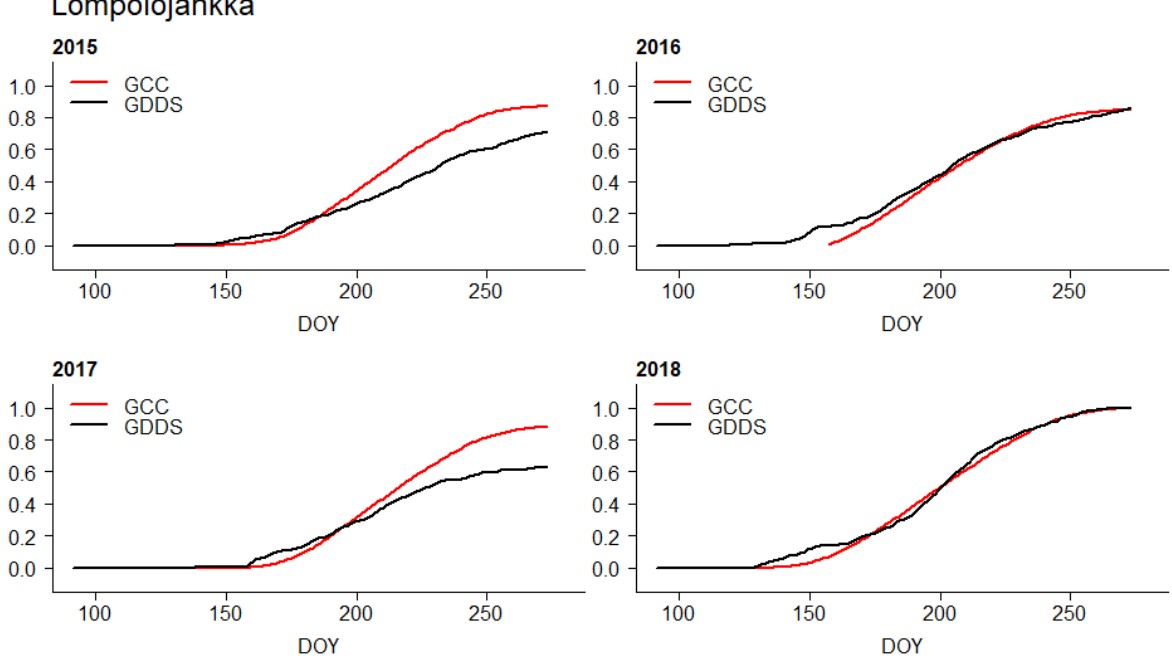

**Figure A4 b: The normalized cumulative GCC and growing degree days (GDDS) in 2015 – 2019 at the experimental sites.**
**The data from 2019 is missing.**

## Kaamanen

**Figure A4 c: The normalized cumulative GCC and growing degree days (GDDS) in 2015 – 2019 at the experimental sites.**





**Figure A5 a: Mean three-day difference in GCC divided to temperature classes (<5 °C, 5-10 °C, >10 °C) at Halssiaapa from May to September. No temperature data in class <5 °C from July and August. The error bars denote the standard error.**



**Figure A5 b: Mean three-day difference in GCC divided to temperature classes (<5 °C, 5-10 °C, >10 °C) at Lompolojänkkä from May to September. No temperature data in class <5 °C from July and August. The error bars denote the standard error.**






**Figure A5 c: Mean three-day difference in GCC divided to temperature classes (<5 °C, 5-10 °C, >10 °C) at Kaamanen from May to September. No temperature data in class <5 °C from July and August. The error bars denote the standard error.**



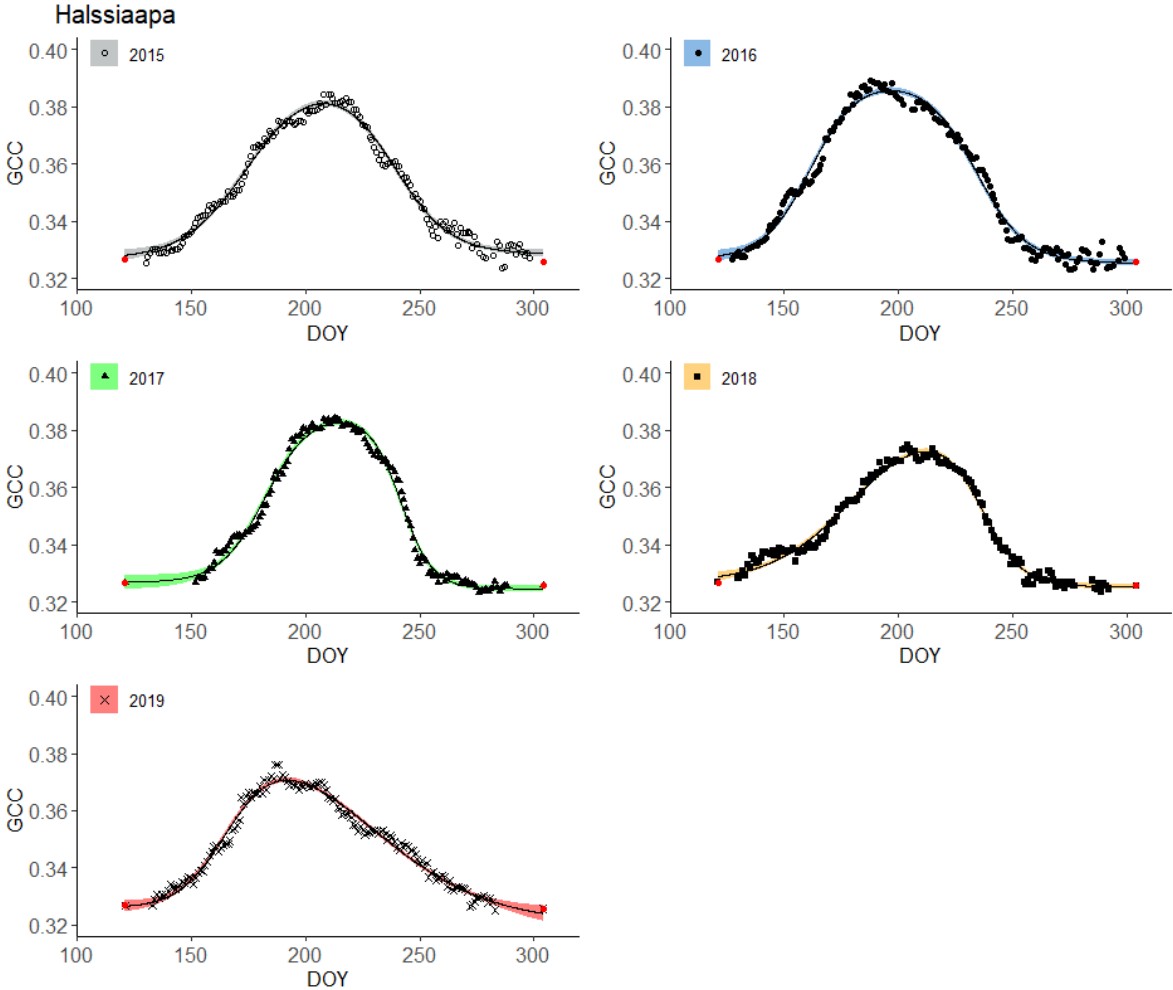

**Figure A6 a: The GCC values and fitted function with 95% confidence intervals in 2015 – 2019 at Halssiaapa. The red dots stand for the fixed start and end point of time in the fitting.**





**Figure A6 b:** The GCC values and fitted function with 95% confidence intervals in 2015 – 2019 at Lomoplojänkkä. The red dots stand for the fixed start and end point of time in the fitting.



**Figure A6 c: The GCC values and fitted function with 95% confidence intervals in 2015 – 2019 at Kaamanen. The red dots stand for the fixed start and end point of time in the fitting.**

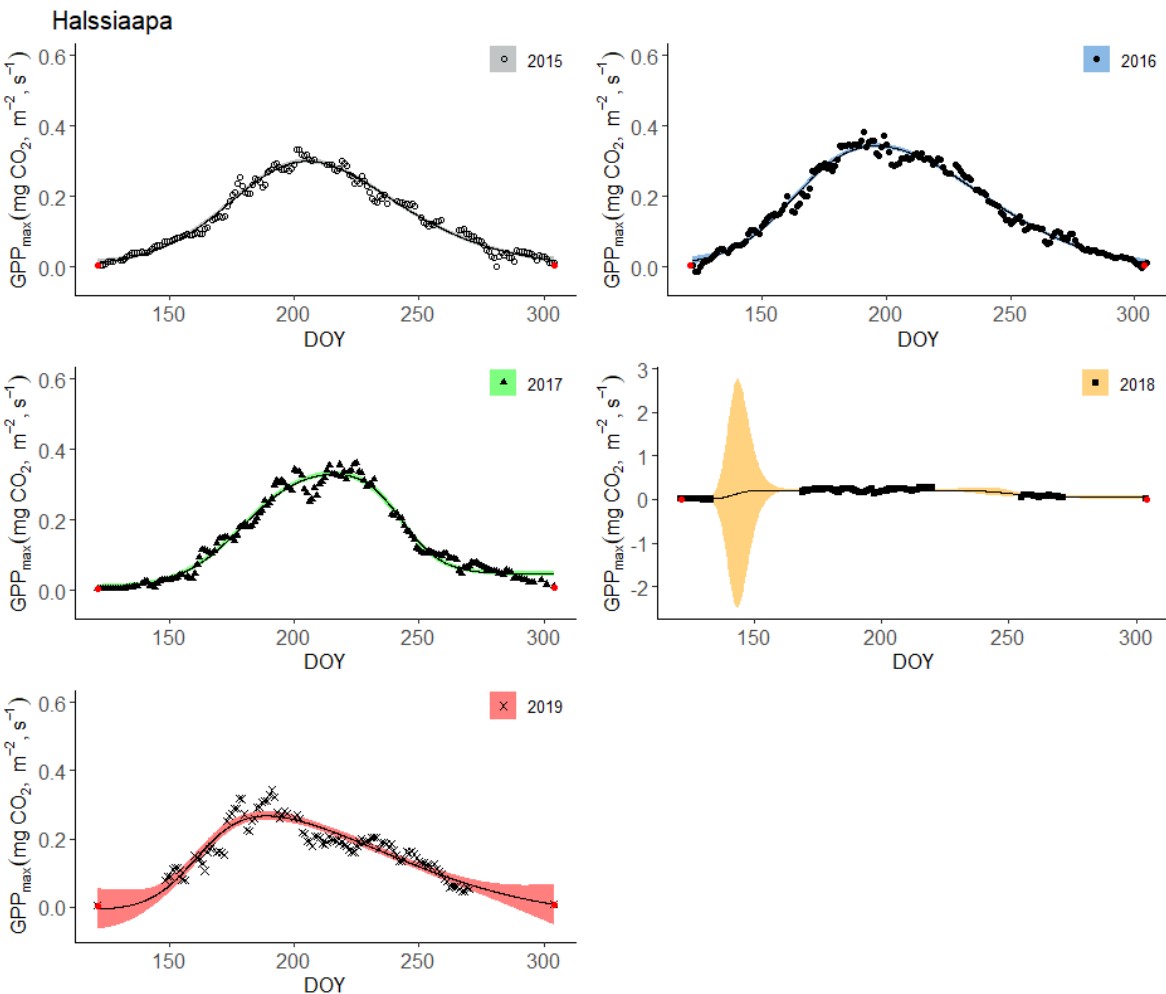

**Figure A7 a: The GPPmax values and fitted function with 95% confidence intervals in 2015 – 2019 at Halssiaapa. Note the different scale in 2018. The red dots stand for the fixed start and end point of time in the fitting.**





**Figure A7 b: The GPPmax values and fitted function with 95% confidence intervals in 2015 – 2019 at Lompolojänkkä. The red dots stand for the fixed start and end point of time in the fitting.**



**Figure A7 c: The GPPmax values and fitted function with 95% confidence intervals in 2015 – 2019 at Kaamanen. The red**
**dots stand for the fixed start and end point of time in the fitting.**





**Figure A8 a: The regression between GCC and GPPmax at Halssiaapa in 2015–2019. The first part of the growing season is denoted with black circles, the latter half with red circles.**



Figure A8 b: The regression between GCC and GPPmax at Lompoljänkkä in 2015–2019. The first part of the growing season is denoted with black circles, the latter half with red circles.



**Figure A8 c: The regression between GCC and GPPmax at Kaamanen in 2015–2019. The first part of the growing season is denoted with black circles, the latter half with red circles.**




Figure A9 a: The scaled GCC and GPPmax data at Halssiaapa in 2015–2019.



**Figure A9 b: The scaled GCC and GPPmax data at Lompolojänkkä in 2015–2019.**



**485**  **Figure A9 c: The scaled GCC and GPPmax data at Kaamanen in 2015–2019.**


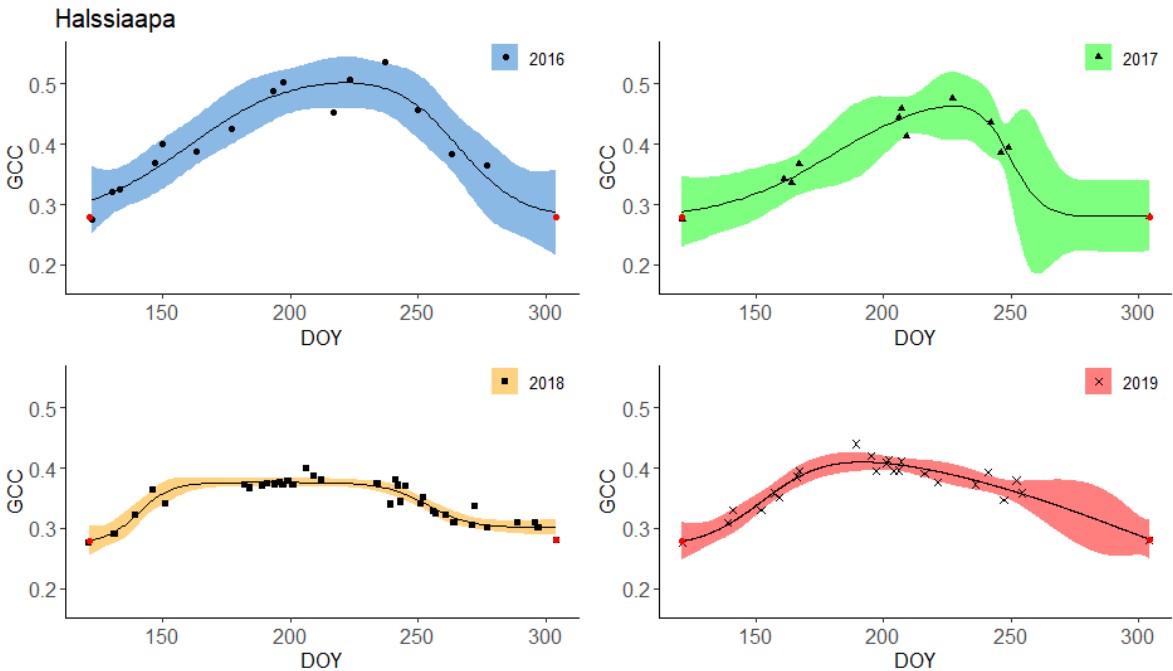

**Figure A10 a: The Sentinel-2 derived GCC values and fitted function with 95% confidence intervals in 2016 – 2019 at Halssiaapa. The red dots stand for the fixed start and end point of time in the fitting.**

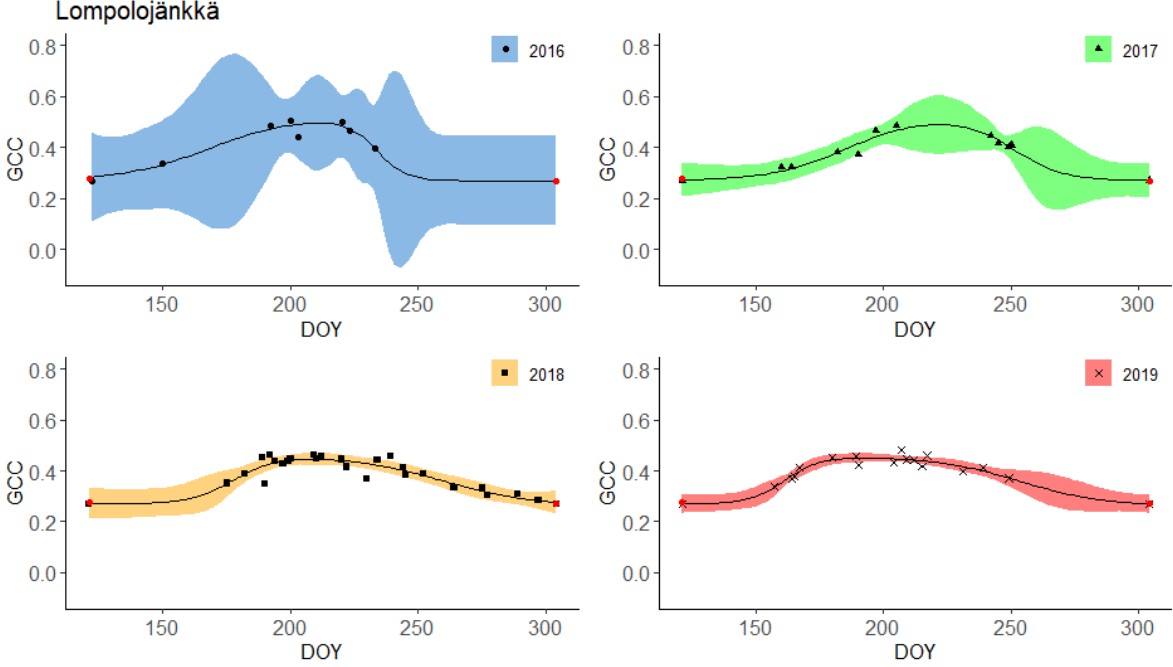

**Figure A10 b: The Sentinel-2 derived GCC values and fitted function with 95% confidence intervals in 2016 – 2019 at Lompolojänkkä. The red dots stand for the fixed start and end point of time in the fitting.**



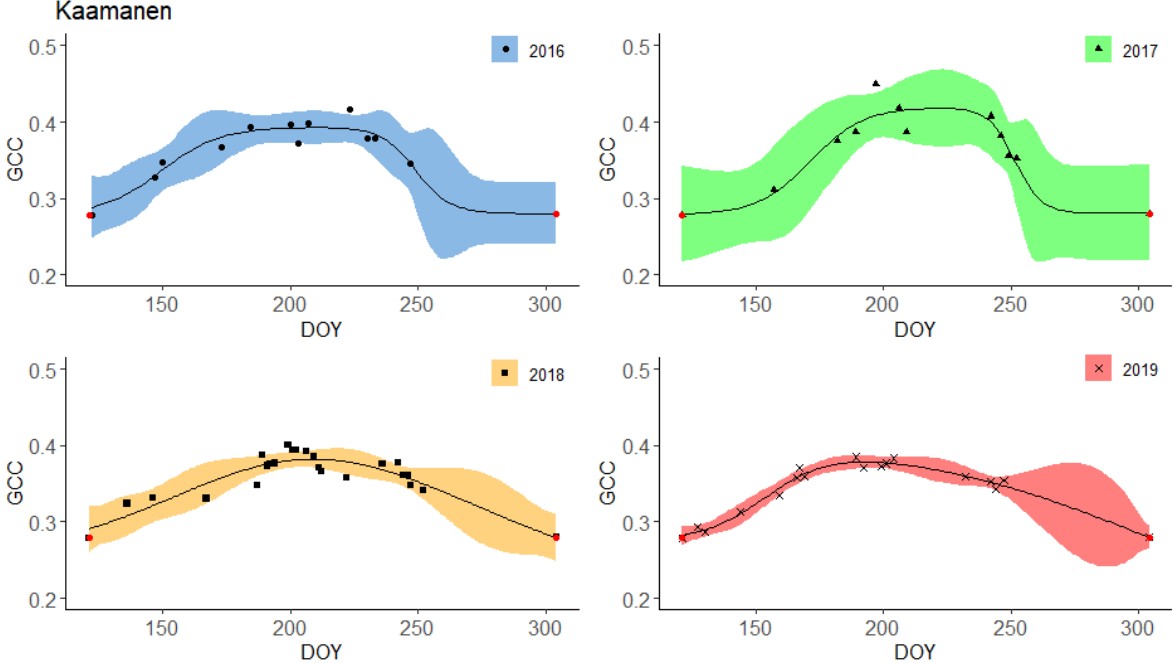

**Figure A10 c: The Sentinel-2 derived GCC values and fitted function with 95% confidence intervals in 2016 – 2019 at Kaamanen. The red dots stand for the fixed start and end point of time in the fitting.**

**Author contribution**

ANA, MP and TL were responsible for enabling and establishing the camera network. MA, ML and JR were part of executing the digital camera setup. MA and AL provided $CO_2$ measurements and environmental variables. MA and J-PT were responsible for post-processing the $CO_2$ data. ON performed the satellite (Sentinel-2) data analysis. CMT developed the image processing tool (FMIPROT). ML performed the digital camera image analysis and drafted the manuscript. MA, ML and J-PT contributed to the

interpretation of results and writing on the first version of the manuscript. All authors commented on the manuscript.

**Competing interests**

The authors declare that they have no conflict of interest.

**Acknowledgements**

This research has been supported by the Academy of Finland (CAPTURE, grant no. 296888) and the EU (MONIMET Project

(LIFE12ENV/FI/000409) funded by EU Life+ Programme 2013–2017).



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
