# Peer review of "Tracking vegetation phenology of pristine northern boreal peatlands by combining digital photography with CO2 flux and remote sensing data"

_Biogeosciences, 2022_

## Referee Comment (RC1)

Review: bg_2022-58

Linkosalmi et al.'s study looked at vegetation phenology at three different boreal peatland sites across 5 different years, linking imagery taken from digital cameras, satellite imagery and productivity data from eddy covariance flux towers. The authors used the green chromatic coordinate to investigate patterns of greenness, and to evaluate the influence of environmental parameters such as temperature and water table level on GCC and subsequent gross primary productivity measurements. I would like to thank the authors for what is a nicely written paper. Overall, it was clear and concise, presenting the data in a coherent and logical manner. You do state you look at water table, but this was not clear to me in your analyses (in fact, it seems missing from the statistical analyses? I would like the authors to expand on how they related GCC and GPP exactly as I felt this was lacking.

Detailed comments:

Line 10: When you state leaf area, do you mean the physical leaf size?

Line 14: I presume you mean air temperature here?

Line 29: Same as abstract, do you mean leaf area as in differences in leaf size?

Line 39: Vegetation phenology is quite a broad concept (incorporating a variety of things such as flowering, leaf out etc). I would maybe include the phrase green leaf phenology here to acknowledge this paper is looking at simply 'greenness'.

Lines 59-63: I was a little confused by the wording here. You state there are three peatlands measured across 5 growing seasons. Your first objective is states however that you would use GCC to describe phenology between sites and among different plant communities at one site. I understand that you are looking at species/community differences within a site, but this makes it sound like you only did this at one site.

Lines 85-101: Did you consider using any other metrics, even other chromatic coordinates such as the Red Chromatic Coordinate?

Lines 195-201: Really nice to show the GCC vales of different species/communities – this is not all that common in the literature, especially for peatlands.

Lines 240-250: This section was a little hard to follow, and I would encourage the author to re-word. It was detail heavy (which is good) but makes it really quite dense. In line 242, you mention a p value, but no other information from the results of the statistical test. I would like more details here. A p value alone is not sufficient when reporting the outputs of statistical analyses.

Figures 7 and 8: Could you maybe incorporate these two figures to show comparison between GCC and GPP more easily? I'd recommend two columns, left hand side GCC, right hand side GPP, with each site in a column (I hope this makes sense). In Figure 8, for Halsiappa, what is the explanation for the yellow shaded area?

Lines 265-268: The wording of the sentence here is quite confusing. So the difference in GCC was significant between Lompolojankka and other two sites in all years, but only significant between Halsiappa and Kaamanen in the last 3 years of study? Think this could be re-worded to be easier to understand.

Figure 9: Your GCC values for some of the sites are really quite high (much higher than other boreal peatland studies such as Peichl et al. 2015 and Davidson et al. 2021). Why is that? Was Halssiappa really that much greener at around Day 240

Line 319: Again, state green leaf phenology here

Line 326: Wording is a little awkward. You state that Lompolojankka is flatter, but the rest of the sentence also makes it sound like there is pronounced microtopography. This should be clearer.

Line 328: 'This affects THE fen's' – word the is missing

Line 330: I'd expanded this section. What might a fen dominated by sedges be doing over say, a bog? Higher productivity in the short term? But faster turnover? Discussion section overall seems rather superficial in areas such as this and could benefit from being expanded upon.

Line 333: How small are the shrubs in this area? Shrubs would typically have a higher GCC unless they are really quite small (thinking Betula spp.?)

Line 336: I wouldn't use the phrase plant growth here, that indicates to me the physical size of the plant, I'd say strictly green leaf phenology.

Lines 340-343: Really cool that you are able to investigate the GCC dynamics of individual plant communities. I would like to highlight another recent paper, investigating GCC dynamics in boreal peatlands, that also looked at community level (finer spatial resolution of 60 x 60 cm) by Davidson et al. 2021:

Davidson, S.J., Goud, E.M., Malhotra, A., Estey, C.O., Korsah, P. and Strack, M. (2021) Linear disturbances shift boreal peatland plant communities toward earlier peak greenness, Journal of Geophysical Research: Biogeosciences https://doi.org/10.1029/2021JG006403

Line 344: This paragraph seemingly comes out of nowhere, and I think it would benefit from having a linking sentence between the previous paragraphs.

Lines 371-372: This sentence is awkwardly worded and I'm not sure I get the meaning. What does phenological courses mean?

Lines 378-380: I may have missed it, but this key information here about the drought in 2018 – could that be included in the study site? I think some more meteorological data would be a really nice inclusion? Show air temperature and precipitation patterns across the 5 year period (and compared to the climate normal)

Conclusion: Unfortunately, I feel like the conclusion lets down what is otherwise a nice paper. I would encourage the others, rather than just summarise the paper here, really place their results within the bigger picture. What can this data help with in the future? There is very little discussion about the usefulness of this type of data beyond explaining productivity patterns at these specific sites. Could this data be used to parameterize models? I don't think you need to go crazy here and go beyond the scope of the paper, but I do think summarising just the paper here weakens the conclusion.

---

## Author Response (AR1)

Response to Referee #1

We thank the reviewer for assessing the manuscript carefully and for the constructive and supportive comments. We have addressed all the comments and questions that were raised (in italics), and the responses are listed below.

In addition to other revisions, the authors decided to move the Appendices to Supplementary material as a separate file due to large number of figures and tables.

*Linkosalmi et al.'s study looked at vegetation phenology at three different boreal peatland sites across 5 different years, linking imagery taken from digital cameras, satellite imagery and productivity data from eddy covariance flux towers. The authors used the green chromatic coordinate to investigate patterns of greenness, and to evaluate the influence of environmental parameters such as temperature and water table level on GCC and subsequent gross primary productivity measurements. I would like to thank the authors for what is a nicely written paper. Overall, it was clear and concise, presenting the data in a coherent and logical manner. You do state you look at water table, but this was not clear to me in your analyses (in fact, it seems missing from the statistical analyses? I would like the authors to expand on how they related GCC and GPP exactly as I felt this was lacking.*
Response: The authors thank the referee for the supportive comments.
We also thank the reviewer for the remark concerning the water table level. The changes in water table level are discussed in relation to changes in temperature in Section 3.2., and we added further discussion about the observed drought. We also revised the aims of the study stated in the introduction.

Detailed comments:

*Line 10: When you state leaf area, do you mean the physical leaf size?*
Response: The term 'leaf area' was changed to 'plant cover'.

*Line 14: I presume you mean air temperature here?*
Response: Yes, this is specified in the revised manuscript.

*Line 29: Same as abstract, do you mean leaf area as in differences in leaf size?*
Response: See above.

*Line 39: Vegetation phenology is quite a broad concept (incorporating a variety of things such as flowering, leaf out etc). I would maybe include the phrase green leaf phenology here to acknowledge this paper is looking at simply 'greenness'.*
Response: Thank you for the suggestion. We clarified the sentence. Also, we checked the whole manuscript for terminology and revised the text where necessary to indicate whether the text refers to greenness specifically or vegetation phenology in general.

*Lines 59-63: I was a little confused by the wording here. You state there are three peatlands measured across 5 growing seasons. Your first objective is states however that you would use GCC to describe phenology between sites and among different plant communities at one*

*site. I understand that you are looking at species/community differences within a site, but this makes it sound like you only did this at one site.*
Response: For clarity, the sentence was corrected from "…within one site…" to "…within each site…"

*Lines 85-101: Did you consider using any other metrics, even other chromatic coordinates such as the Red Chromatic Coordinate?*
Response: The GCC has previously proven to be a valid metric for greenness in peatlands (e.g. Peichl et al. 2015, Linkosalmi et al. 2016, Koebsch et al. 2019), so we decided to use it instead of other indices. The RCC would certainly be an interesting metric, too, for peatland vegetation, but here we wanted focus on the greenness and, in order to keep the amount of data within reasonable limits, the RCC was not included in the analysis.

*Lines 195-201: Really nice to show the GCC vales of different species/communities – this is not all that common in the literature, especially for peatlands.*
Response: We thank the referee for the supportive comment.

*Lines 240-250: This section was a little hard to follow, and I would encourage the author to re-word. It was detail heavy (which is good) but makes it really quite dense. In line 242, you mention a p value, but no other information from the results of the statistical test. I would like more details here. A p value alone is not sufficient when reporting the outputs of statistical analyses.*
Response: We made some editorial changes to improve the clarity of this paragraph. The p-value in question did not actually refer to any specific statistical test and it was removed. In other Sections, the $chi^2$ values were added to report the results of statistical analysis, in addition to the p-values.

*Figures 7 and 8: Could you maybe incorporate these two figures to show comparison between GCC and GPP more easily? I'd recommend two columns, left hand side GCC, right hand side GPP, with each site in a column (I hope this makes sense). In Figure 8, for Halsiappa, what is the explanation for the yellow shaded area?*
Response: We thank the referee for the suggestion, which was implemented. The explanation for the yellow shaded area (wide confidence interval due to a data gap) at Halssiaapa in 2018 was explained in the results (Section 3.3) and discussion.

*Lines 265-268: The wording of the sentence here is quite confusing. So the difference in GCC was significant between Lompolojankka and other two sites in all years, but only significant between Halsiappa and Kaamanen in the last 3 years of study? Think this could be re-worded to be easier to understand.*
Response: The sentence was re-worded for clarity.

*Figure 9: Your GCC values for some of the sites are really quite high (much higher than other boreal peatland studies such as Peichl et al. 2015 and Davidson et al. 2021). Why is that? Was Halssiappa really that much greener at around Day 240*
Response: The values in Fig. 9 represent the satellite derived GCCs, which are higher than the camera derived GCCs (Fig. 7). Peichl et al. (2015) and Davidson et al. (2021) used camera derived indices and their result are in accordance with our data. In discussion, we note that

the GCC values differ between satellite and camera derived images and discuss the possible reasons for this ("...the different viewing angles and atmospheric effect (the scattering and absorption of radiation due to atmospheric molecules and aerosols) and the consequent atmospheric correction of the satellite data"). Also, the model fit is still passable, although there is statistical scatter in the data and the point at day 240 represents the highest value.

*Line 319: Again, state green leaf phenology here*
Response: Corrected

*Line 326: Wording is a little awkward. You state that Lompolojankka is flatter, but the rest of the sentence also makes it sound like there is pronounced microtopography. This should be clearer.*
Response: The sentence was edited. The pronounced microtopography refers to the other two sites (Halssiaapa and Kaamanen)

*Line 328: 'This affects THE fen's' – word the is missing*
Response: Corrected

*Line 330: I'd expanded this section. What might a fen dominated by sedges be doing over say, a bog? Higher productivity in the short term? But faster turnover? Discussion section overall seems rather superficial in areas such as this and could benefit from being expanded upon.*
Response: The discussion was reworded. The authors meant to discuss the special characteristics of this specific fen, Lompolojänkkä, rather than the general differences between fens and bogs.

*Line 333: How small are the shrubs in this area? Shrubs would typically have a higher GCC unless they are really quite small (thinking Betula spp.?)*
Response: The dominant plants that defined the ROI selection at the sites are presented in Table 1. 'Shrubs' refer to species (such as Andromeda polifolia, Empetrum nigrum at Halssiaapa and Kaamanen) smaller than Betula spp., which we have separated as its own ROI. The text was edited and clarified, as greater GCC values were found in general for annuals (such as Menyanthes trifoliata) and taller woody plants (such as Salix spp. and Betula spp.) than for smaller shrubs and sedges, but at Kaamanen the shrubs had a higher GCC than the sedges and mosses.

*Line 336: I wouldn't use the phrase plant growth here, that indicates to me the physical size of the plant, I'd say strictly green leaf phenology.*
Response: Corrected

*Lines 340-343: Really cool that you are able to investigate the GCC dynamics of individual plant communities. I would like to highlight another recent paper, investigating GCC dynamics in boreal peatlands, that also looked at community level (finer spatial resolution of 60 x 60 cm) by Davidson et al. 2021: Davidson, S.J., Goud, E.M., Malhotra, A., Estey, C.O., Korsah, P. and Strack, M. (2021) Linear disturbances shift boreal peatland plant communities toward earlier peak greenness, Journal of Geophysical Research: Biogeosciences https://doi.org/10.1029/2021JG006403*

Response: We thank for the supportive comment and for the reference, which was added to discussion.

*Line 344: This paragraph seemingly comes out of nowhere, and I think it would benefit from having a linking sentence between the previous paragraphs.*
Response: The previous paragraph was edited to provide a conceptual link.

*Lines 371-372: This sentence is awkwardly worded and I'm not sure I get the meaning. What does phenological courses mean?*
Response: The sentence was edited and "phenological courses" was changed to "phenological development".

*Lines 378-380: I may have missed it, but this key information here about the drought in 2018 – could that be included in the study site? I think some more meteorological data would be a really nice inclusion? Show air temperature and precipitation patterns across the 5 year period (and compared to the climate normal)*
Response: We thank for the suggestion. We included a table to show monthly air temperatures and precipitation sums. It was referred to in the description of the study sites as well as in Results and Discussion.

*Conclusion: Unfortunately, I feel like the conclusion lets down what is otherwise a nice paper. I would encourage the others, rather than just summarise the paper here, really place their results within the bigger picture. What can this data help with in the future? There is very little discussion about the usefulness of this type of data beyond explaining productivity patterns at these specific sites. Could this data be used to parameterize models? I don't think you need to go crazy here and go beyond the scope of the paper, but I do think summarising just the paper here weakens the conclusion.*
Response: We thank the reviewer for the comment. We elaborated the conclusions and added discussion about the use of the data for monitoring and modelling of ecosystems.

We thank the reviewer for assessing the manuscript carefully and for the constructive and supportive comments. We have addressed all the comments and questions that were raised (in italics), and the responses are listed below.

In addition to other revisions, the authors decided to move the Appendices to Supplementary material as a separate file due to large number of figures and tables.

*The current manuscript uses digital repeat photography for three peatland sites and compares this to other seasonally-variant observations, i.e. on precipitation, temperature, CO2 fluxes, and Sentinel-2 reflectance. The manuscript was clearly written, was scientifically sound, and provides interesting analyses. Nonetheless, I have a few considerations that may help to improve the work. My main comments are:*

*1. Some of the methodological choices are not sufficiently justified and evaluated against other approaches. For example, L100 indicates that daily GCC averages were used. However, with similar input data many reference studies use the 90th percentile of GCC values over 3 days (e.g.: https://doi.org/10.1016/j.agrformet.2011.09.009 ) (SOnnentag), which may help to reduce illumination effects and consequently smoothen the GCC profiles: why was this approach not followed here? Another question is why a 25% threshold for SOS/EOS was chosen (L166-167): although not an uncommon value, it would require justification. (see further issues in "specific comments")*

Response: We have previously (Linkosalmi et al., 2016, (https://doi.org/10.5194/gi-5-417-2016)) used the daily GCC averages after determining the optimal time window for deriving the GCC from the images, based on an analysis of grey reference plates. Also, evaluation of the daily GCC values enables even the detection of rapid changes in the GCC values. This was further emphasized in the text. The 25% threshold is in accordance with Richardson et al. (2018) (https://doi.org/10.1038/sdata.2018.28). In addition, at peatland sites the probability of excess surface water after the snowmelt is high, and thus a relatively high threshold is preferable to reduce the effect of water surfaces. We added an explanation to the revised manuscript.

*2. While the EC method is an important reference, it remains unclear how the EC footprint relates to the various ROIs and how it can thus effectively represent the variability of CO2 exchange as caused by the different vegetation elements (and how this relates to the analysis provided here). We note that to better describe the phenology around AmeriFlux/NEON tower footprints, recently a paper was submitted to Scientific Reports that uses 3-m resolution Planet data to extract 10x10 km phenology for each EC tower.*

Response: It is clear that the EC measurements represent an area larger than an individual plant community area and that the vegetation varies within the flux footprint. However, we related the $CO_2$ fluxes to the general ROI (including many plant communities), not to the smaller ROIs defined for specific plant community types.

*3. L339-342: "to our knowledge" may require some careful checking of literature also, although "with this precision" gives room for interpretation. It would be better if the authors could relate to other studies that also separate different elements in phenoCams. Examples exist (although "with this precision" may need to be clarified!):*
*https://doi.org/10.1016/j.rse.2020.112004 take different parts of the phenoCam image to look at grass/shrub/tree signals in a savanna.*
*https://doi.org/10.1016/j.agrformet.2014.08.007 divides the camera image into small subsets for which SOS is calculated. https://doi.org/10.3389/fpls.2015.00110 looks at individual tree crowns in a single image. I would expect other studies to do this too. Perhaps this is not what authors mean, but I'd highly recommend to expand the "to our knowledge" to better clarify the innovation here and put it in perspective.*

Response: We agree. The text was edited and references were added.

Specific comments:

*- L32-41: the authors correctly indicate that a change of abiotic conditions (particularly warming) affects the C-balance due to increased take up of CO2. While this is correct, warming in peatlands also causes high CO2 emissions. While not a topic in this study, this aspect of the carbon balance could be highlighted here.*
Response: The authors thank the referee for the suggestion and added a sentence about the effect of warming on the peat soil C losses.

*- L86-87: could the authors also indicate the height on the pole where the camera was mounted? This is quite crucial information in my view. The reported angle is probably the depression angle?*
Response: The angles and heights of the cameras were re-estimated and the text was updated to include this information.

*- Figure 1: red lines and numbers on a green background are not very clear. Particularly also for 10% of male who are red/green colorblind. I suggest changing color and increase siz of the numbers.*
Response: The colour palette was changed to more a colourblind-friendly one and the font sizes were increased.

*- L126: why was a base temperature of 5 degrees used here, and not 0 degrees for example? Could authors provide justification for this in the manuscript?*
Response: In a sense, the choice of the base temperature is arbitrary. Here we followed the convention adopted by the Finnish Meteorological Institute (FMI). This explanation was added to the text.

*- L129-131: I would request the authors to rewrite these two sentences: I could not understand it. "Monthly average" of what and how can an average be divided in 3? What is the "value just before the increase"?*
Response: The sentences were edited to improve clarity.

*- L145-146: the minimum of two days is because there are overlapping orbits: this should be*

*mentioned. Also I would like to read about how many cloud-free observations were available on average.*
Response: The sentences were rephrased.

*- L183-184: why not 15th, but 17th of June in Kaamanen?*
Response: After checking the image directory again, we found an image from Kaamanen for 15 June 2015. The image was changed and the figure caption was corrected.

*- Table 2: please indicate in caption why those data are missing. In addition, explain why some entries are in bold font. Possibly the highest/lowest numbers? But then by for Lompolojänkka there are two (different) bold values for Max GP week?*
Response: The caption was edited and the bolded values were corrected to the table.

*- L200: could somehow the significance of these differences be indicated in the table?*
Response: The significance of these differences is presented in Table A2, which is referred to in the text.

*- Figure 6: the figure now suggests that Lom for June > 10C is not significantly different from the others? Just to be sure that I interpret correctly, because the error bars suggest no overlap with the other two.*
Response: The authors thank for the remark. Mistakes in the figure were corrected. In addition, the significant differences were added to Tables S5 & S6 in Supplementary material.

*- Table 4: I could not find a clear explanation for the low R2 of 2018a at Halssiaapa: or is this because of what is written in L260-261?*
Response: The explanation was added to Section 3.3.

*- L371-375: great that the authors manage to also use the GCC levels; this is probably because the StarDot is a stable camera, whereas for cheap cameras (such as in https://doi.org/10.1016/j.jag.2020.102291) this is less the case.*
Response: The authors agree and thank for the comment.

*- L394-396: this seems a relevant discussion. I suppose that the authors imply that for the vegetation that they study less of such non-photosynthetic biomass is present? In addition, the depression angle used by Vrieling et al (2018) is much smaller (i.e. less towards nadir) than in this study.*
Response: The authors did not mean to take a stance on the amount of non-photosynthetic biomass, discussing instead the differences between camera and satellite data derived phenological phases. We rephrased the text for clarity.

*- L401: please specify "typically" every 5 to 10 days is for Sentinel-2 in general without overlapping orbits, but not for non-cloudy satellite images.*
Response: The sentence was rephrased to clarify the image availability.

*- L405-409: in this framework the RS mapping with PlanetScope could also be mentioned;*

*several efforts exist at present, and the satellite constellation offers very frequent imagery at fine spatial resolution (3m).*
Response: The text was edited. We added a reference to PlanetScope.

*- L419-420: this statement is a bit vague "more satellite data would be needed". The authors probably mean a finer temporal resolution resulting in more frequent cloud-free observations? Again, see also the previous comment.*
Response: The sentence was improved for clarity.

*- General: are the camera-data and/or GCC data somewhere available on a repository and/or part of a network like https://phenocam.sr.unh.edu?*
Response: Unfortunately not at the moment

*- Figure A2: WTD is missing for 2019? Please report why in caption.*
Response: The explanation was added to the caption.

*- Figure A3: GGDS: S is for "sum"? Add to caption for clarity.*
Response: Thank you. We corrected the caption.

*- Figure A5b: caption: I believe that only no temperature data in class <5 for August (July should be deleted here).*
Response: "July" was deleted.

Edits:

*- L40: "has been verified" is somewhat vague here: could authors be more specific on the findings of those studies?*
Response: We added a brief description of these findings.

*- L89: "in all cameras" should read "for all cameras"*
Response: Corrected.

*- L114: "on an" should read "at the"*
Response: Corrected.

*- L139: "filtered", but also "discarded" in the subsequent analysis?*
Response: The sentence was edited.

*- L351: remove "those" and replace "which" with "that"*
Response: Corrected.

---

## Author Response (AR2)

Response to Referee #1

We thank the reviewer for reassessing the manuscript, for the corrections and for the supportive comments. We have addressed the comments and remarks (in italics), and the responses are listed below.

*I thank the authors for putting the work in and taking on board my suggestions. It has definitely improved the manuscript immensely.*

*Minor comments:*
*Line 389: I would provide latin name for big-leaved bog bean here for clarity - you use latin names everywhere else.*

We thank for the suggestion, the name was changed to latin.

*Line 399: The word decomposing is awkward in this context. I suggest changing it*

The word was changed to "partitioning".

*Lines 405-406: Awkward sentence structure.*

The authors thank for the remark, the sentence was modified.

*Line 412: Change to 'Our results imply that northern peatland vegetation is capable of quick growth following a cold spring'*

The authors thank for the suggestion, the sentence was modified.

We thank the reviewer for reassessing the manuscript, for the corrections and for the supportive comments. We have addressed the comments and remarks (in italics), and the responses are listed below.

*I thank the authors for their responses and adaptations to the manuscript. I have read the revised manuscript again. In my opinion it is ready for publication, and only have the following minor observations (line numbers relate to track changes version):*
*- L41: "verified" is a strange word to use here. Rather something like: "various studies have found a strong relationship between …" (in fact, I would combine this sentence with the next and write a single clear point).*

The authors thank for the suggestion, the sentences were modified and combined.

*- L44: "has been suggested" is vague. I believe that based on many studies we can say that it "is" a key driver?*

The sentence was modified and "has been suggested" was corrected as "acts".

*- Although I read the response to my general comment #2 in the previous round, looking at Figure 1 I still wonder how these ROIs (even if here the "general ROI" was used) may relate to the EC footprint. For example, on Figure 1 and 2 I see trees (forest) around the Lompolojänkkä site as well (and in fact even nearby branches of the Betula tree, which is not in the "general" ROI of Figure 1b). Possibly the authors could provide a little more confidence to the readers in explaining how the ROI links to the footprint, for example by adding something in Section 2.4 on the heights of the flux towers, and predominant wind direction.*

More information about the flux measurement systems and data processing was added to Section 2.4, including the measurement heights and filtering of the flux data. These amendments indicate that the measurement data accepted for the present analysis are not significantly affected by surrounding forests, i.e., at each site, the flux footprint predominantly covers the open peatland area around the flux tower, which coincides with the target area included in the general (larger) ROIs.

*- L163: up to editor also, but usually in this case numbers until 10 are written as words in scientific text: "five to ten days".*

The authors thank for the remark, this was corrected.

*- L193: "eliminated" or "reduced"? "Differences" or "derivatives"?*

"Eliminated" was changed to "taken into account", but indeed differences were used in the transformation.

*- Table 2: are those p-values correct? I would expect a minus sign in the exponent? Now these are*

*very large values… Rather than such precise numbers, the authors could also indicate commonly used p threshold levels like <0.001 (which authors also use later in text to indicate significant differences in GCC).*

The authors thank for the remark and suggestion, the p-values were corrected.

*- L232/255/258/267 and elsewhere: I think authors can leave out "material". Just "Supplementary Table S3 and S4". Check with other papers in this journal.*

The authors thank for the remark, these were corrected.

*- L422: "of" should read "by"*

The authors thank for the remark, this was corrected.

*- L475/477: "oblique" instead of "angled"?*

The authors thank for the suggestion, this was corrected.

*- L520: "input" instead of "material"?*

The authors thank for the suggestion, this was corrected.